# A barley pan-transcriptome reveals layers of genotype-dependent transcriptional complexity

A pan-transcriptome describes the transcriptional and post-transcriptional consequences of genome diversity from multiple individuals within a species. We developed a barley pan-transcriptome using 20 inbred genotypes representing domesticated barley diversity by generating and analyzing short- and long-read RNA-sequencing datasets from multiple tissues. To overcome single reference bias in transcript quantification, we constructed genotype-specific reference transcript datasets (RTDs) and integrated these into a linear pan-genome framework to create a pan-RTD, allowing transcript categorization as core, shell or cloud. Focusing on the core (expressed in all genotypes), we observed significant transcript abundance variation among tissues and between genotypes driven partly by RNA processing, gene copy number, structural rearrangements and conservation of promotor motifs. Network analyses revealed conserved co-expression module::tissue correlations and frequent functional diversification. To complement the pan-transcriptome, we constructed a comprehensive cultivar (cv.) Morex gene-expression atlas and illustrate how these combined datasets can be used to guide biological inquiry.

Barley is a highly adaptable cereal crop that underpins key food, feed and drink sectors across the world[1]. Its diploid inbreeding genetics have led to it being considered a model for more genetically complex temperate small grain cereals of the *Triticeae* tribe, which include hexaploid, tetraploid and diploid wheat, outbreeding rye and synthetic *Triticale*[2,3]. A blend of gene-centric and comparative genomic studies has demonstrated that barley's morphological and developmental plasticity and general adaptability are, in many cases, the result of natural or induced genetic variants that enhance reproductive success in diverse agroecological environments[4–7]. Recently, the type and scale of genetic variation in the barley genome were revealed in detail through a comparative analysis of chromosome-level sequence assemblies from diverse genotypes that are representative of the species' global diversity space[8–10]. The resulting 'pan-genome' revealed the impact of both natural and induced postdomestication mutations on genome integrity and the origin of phenotypic variants selected and maintained over time through human selection[11]. While these data

are proving exceptionally powerful, the broader consequences of the observed genome diversity have not yet been assessed. Here we explore the functional implications of genotypic diversity by characterizing the landscape of transcriptional variation within and between the barley genotypes used to construct the V1.0 pan-genome[8].

## Results

### A barley pan-transcriptome

To construct a barley pan-transcriptome, we performed RNA-sequencing (RNA-seq) and PacBio Iso-seq on three replicates of five diverse tissues (Fig. 1a) isolated from the 20 genotypes representing V1.0 of the barley pan-genome[8]. We identified 'tissue' as the main driver of transcript diversity (Fig. 1b). To avoid genotypic bias introduced by using a single reference for mapping RNA-seq reads[12], we produced genotype-specific reference transcript datasets (GsRTDs) for each of the 20 individuals. GsRTDs assemble a broader range of transcripts than a single reference, producing more comprehensive

✉e-mail: micha.bayer@hutton.ac.uk; craig.simpson@hutton.ac.uk; runxuan.zhang@hutton.ac.uk; robbie.waugh@hutton.ac.uk

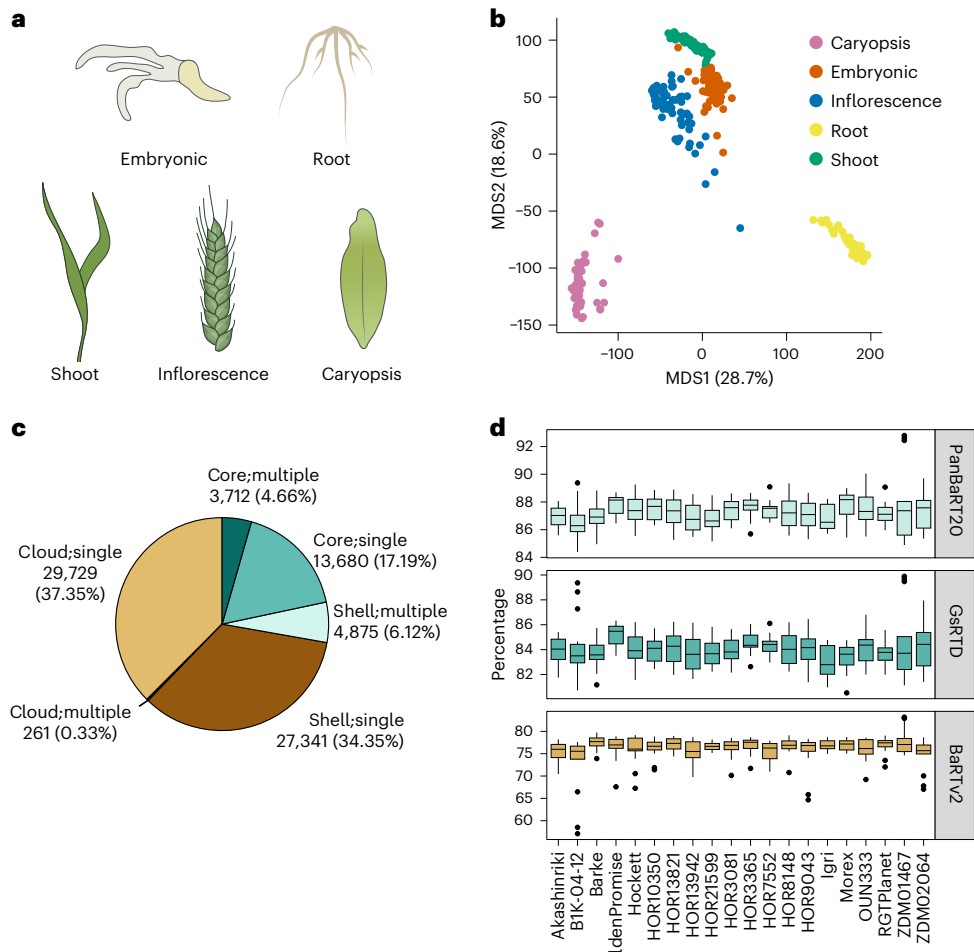

**Fig. 1 | Transcript diversity and classification. a,** Five tissues sampled (clockwise from top left) are as follows: embryo, mesocotyl and seminal roots (from here referred to as 'embryonic' tissue), seedling root, caryopsis, developing inflorescence and seedling shoot. **b,** An MDS plot of the overall transcript abundance data. Different colors represent different tissues as indicated. **c,** The percentage of different gene classifications (core, shell and cloud genes with single or multiple copies) in PanBaRT20. **d,** Average mapping rates of 14 (HOR8148) and 15 (remaining genotypes) samples for the RNA-seq data from different cultivars to the three transcriptomes (PanBaRT20, respective GsRTDs and BaRTv2). The lines within the boxes display the median, with the box bounds representing the 25th and 75th percentiles. The whiskers extend to the minimum and maximum values within the 1.5× interquartile range. Data points outside this range are outliers.

and accurate transcriptomes that improve expression analyses[13–16]. The number of genes in the 20 GsRTDs ranged from 35,500 to 40,800, with an average of 38,400. They revealed an average of 3.22 transcripts per gene, significantly higher than traditional genome annotations. The cultivar (cv.) Barke GsRTD (GsRTD$_{Barke}$) comprised 39,200 genes and 138,600 transcripts with an average of 3.54 transcripts per gene, closely resembling the recently published cv. Barke reference transcript dataset (RTD; BaRTv2.0 (ref. 16)) assembled from a broad set of tissues, treatments and a reference genome (a detailed comparison of barley Illumina and PacBio transcript assemblies is described in more detail in ref. 16). These statistics reflect the high quality of the assembled GsRTDs (Supplementary Data 1).

We explored the possibility of generating a unified transcriptome for domesticated barley, a pan-RTD, as a reference for RNA-seq analyses from diverse genotypes. We constructed a linear pan-genome using Pan-genome Construction and Population Structure Variation Calling Pipeline (PSVCP)[17] by integrating the 20 reference barley pan-genomes[8] using cv. Morex as the backbone (Supplementary Data 2) and then mapping and clustering orthologous transcripts from the 20 GsRTDs onto this framework (Methods; Extended Data Fig. 1). Clustering transcripts on a linear genome allowed us to directly compare features, including gene structures, alternative splicing,

transcription start and stop sites and copy number variation (CNV) across all 20 genotypes. The resulting pan-RTD, PanBaRT20, includes 79,600 genes and 582,000 transcripts, with a diversity of 7.3 transcripts per gene. The GsRTD$_{Morex}$ provided exceptional mapping success with only 32 genes (0.08%) unmapped, likely due to cv. Morex being the 'backbone' of the linear pan-genome reference. In contrast, on average 9.81% of the genes in a GsRTD were not present in Pan-BaRT20 (Extended Data Fig. 2a).

Mapping the GsRTDs onto the linear pan-genome identifies relationships between GsRTDs and PanBaRT20 directly (Methods). Thus, 13,700 genes in PanBaRT20 mapped uniquely to a single locus in all 20 GsRTDs. We denote these as core, single-copy genes (core;single). These accounted for 17.19% of the PanBaRT20 genes and, on average, 35.7% of the genes in each GsRTD (Fig. 1c). PanBaRT20 also contained 3,700 (4.66%) core genes with multiple matched locations (core; multiple). PanBaRT20 genes found in 2–19 GsRTDs were defined as shell genes that could be further partitioned into single copies (27,300 genes) and multiple copies (4,900 genes). The remaining 30,000 (37.69%) genes were designated as cloud genes, matching a single GsRTD (Supplementary Data 3). Gene ontology (GO) enrichment[18] revealed that core genes were associated largely with ubiquitous biological functions (for example, DNA, nucleus, nucleolus

and transcription), while responses to biotic and abiotic stresses dominated the shell and cloud gene categories (for example, 'defense response' and 'response to stimulus'; Extended Data Fig. 2b), consistent with observations on the pan-genome[8,9].

## GsRTDs and PanBaRT20 as references for transcriptome analyses

A key argument for developing a pan-genome is that it reduces bias in genomic analysis when compared to a single common reference. We therefore investigated how different reference transcriptomes would bias quantification accuracy in RNA-seq datasets. Analyzing the genotype-specific transcriptomic data against PanBaRT20 revealed an average mapping rate of 87.3%, 11.1% higher than BaRTv2.0 (ref. 16; 76.2%), but only 3.3% higher than the average mapping rates of GsRTDs (84.0%; Fig. 1d). To explore further, we simulated RNA-seq reads from five diverse genotypes (Akashinriki, Barke, Golden Promise, Morex and OUN333) and analyzed them with PanBaRT20, individual GsRTDs and GsRTD$_{Morex}$, which serves as a proxy for a common reference (Methods). We found that the respective GsRTDs achieved the highest quantification accuracy, with both PanBaRT20 and GsRTDs outperforming GsRTD$_{Morex}$. GsRTDs, when available, are therefore best for RNA-seq read quantification, but a pan-RTD is generally more appropriate than a single common reference.

## Evidence-based pan-genome annotation

A study discussed in ref. 8 previously generated reference-quality genome assemblies from the same 20 genotypes used here, but transcript-based evidence from only three genotypes was used to annotate the remaining 17 via a process of consolidation and projection. Interestingly, the number of gene models was about 20% higher in the projections than in de novo annotations. An important application of the pan-transcriptome data has therefore been its value in evidence-based re-annotation of all 20 pan-genome assemblies (as described in an updated pan-genome study[9]). This expanded set of gene models with transcriptional evidence was once again consolidated and projected onto an extended set of 76 pan-genome genotypes, enhancing the overall value of the V2.0 pan-genome.

## Drivers of transcript abundance variation

PanBaRT20 revealed a substantial increase in the number of transcripts per gene over the GsRTDs (from 3.5 to 7.3). This is reflected in an increased number of nonredundant splice junctions (SJs) from an average of 146,600 in a GsRTD to 311,300 in PanBaRT20. To illustrate, within the highly expressed transcripts of core single-copy genes (average transcripts per million (TPM) >10), 132 retained introns, 112 had alternative 3' splice sites, 66 had alternative 5' splice sites, 17 had skipped exons and 17 had alternative first exon events, variations generating unique genotype-specific transcripts in the pan-transcriptome. For example, both chr2H11235 (encoding a DNA helicase) and chr3H26163 (unknown function) have 5' splice site mutations that abolish specific splice sites in 7 and 11 of the 20 genotypes, respectively. For chr2H11235, the mutation leads to activation of an alternative 5' splice site 12 nucleotides (nt) upstream with the loss of four amino acids in seven genotypes. For chr3H26163, the mutation results in an intron retention (Fig. 2a and Supplementary Data 4). Considering alternative transcript abundances, in caryopsis over half of the transcripts from chr5H50838 (encoding a formate tetrahydrofolate ligase ortholog) are a single 2,842 nt isoform containing a 721 amino acids open reading frame (ORF). This transcript is replaced by a range of different transcripts in inflorescence and root tissues but mostly by a 2,326 nt transcript that loses 219 amino acids from the N terminus of the protein due to an alternative transcription start. This shorter transcript also has an alternative 5' splice site that generates an upstream ORF before the main protein-coding sequence (CDS). In all cases, the observed variants may affect the translation and function of the protein products.

We next explored the impact of presence/absence (PAV) and CNV on transcript abundance. Across the 20 GsRTDs, 2,925 genes exhibited zero TPM in all tissues in 1–19 genotypes (Fig. 2b). In 2,899 of these cases, their respective genes were absent in the pan-genome assemblies and could therefore be classified as PAVs. They are enriched for GO terms associated with 'modulation' and 'response', which we interpret as them being potentially associated with 'conditional dispensability'. We then used cluster analysis on PanBaRT20 to explore the extent of CNV across the pan-genome. We identified 723 tandem gene clusters, with 98 showing a significant positive correlation ($r > 0.45$ and $P < 0.05$) between copy number and transcript abundance in one or more tissues (Fig. 2c and Supplementary Data 5). As an example, CNV of a 17 kb segment on chromosome 5H harboring cold-induced C-repeat/DRE-binding factor (CBF) 2a (chr5H52041) and 4b (chr5H52037) genes has previously been shown to be correlated with frost tolerance[19]. Our data revealed one to five copies of both HvCBF2a and HvCBF4b in the 20 pan-transcriptome genotypes. Despite there being no cold treatment in our biological samples, the higher copy number did, however, exhibit higher basal gene expression (Extended Data Fig. 3a and Supplementary Data 6). As transgenic experiments show that overexpression of CBFs induces 'cold response genes' in the absence of cold[20], we speculate that high transcript abundance as a consequence of CNV may thus provide sufficient basal CBF gene expression to prime low-temperature responses. Accordingly, three of the six winter growth habit genotypes show high CNV in comparison to only 1 of 14 spring types containing more than one copy.

The original barley pan-genome[8] revealed a 141 Mb inversion on chromosome 7H in a single genotype (cv. RGT Planet). To assess the potential functional impact of this inversion, we interrogated a gene-expression quantitative trait locus (eQTL) dataset comprising RNA-seq data from four tissues of 201 contemporary spring barley cultivars, including cv. RGT Planet. The eQTL analysis clearly identified 31 genotypes containing the inversion and 170 without[21] (Supplementary Data 7). Within the inversion, we observed 36 significantly upregulated and 39 significantly downregulated ($-0.5 < \log(\text{fold change (FC)}) > 0.5$; $\log(\text{adjusted } P) > 10$) genes (Extended Data Fig. 3b,c and Supplementary Data 8). Intriguingly, all upregulated genes were located near the telomeric end of the inversion and all downregulated genes toward the centromere (Fig. 2d). This region has been previously associated with grain-size-related traits, starch content, protein content and α-amylase activity[22–27]. Of the starch metabolism genes, α-glucosidase 1 (chr7H77818; HvAgl1), an α-amylase 2 cluster (chr7H76138, chr7H76155 and chr7H76154; HvAmy2) and starch-branching enzyme 3 (chr7H75382; HvSbe3) are all found within the inversion. HvSbe3 is significantly lower expressed in the lines containing the inversion (Supplementary Data 8). Moreover, 57 of the differentially expressed genes (DEGs) are expressed in a developing barley grain series indicating the importance of this region for grain-related traits[28]. Given the size of the inversion, we hypothesized that the observed differential expression was a consequence of switching the physical location of the respective genes between transcriptionally permissive and repressive nuclear compartments. Supporting this hypothesis, principal component analysis (PCA) of high-throughput chromosome conformation capture (Hi-C) interaction data[10], which partitioned the genome into active (A) and inactive (B) compartments, revealed that the 7H inversion encompassed the A/B compartment boundary (Extended Data Fig. 3d). This genomic region is particularly relevant for breeding as it will not recombine. Of note, within the eQTL dataset, the inversion is now present in 40% of accessions registered in the United Kingdom since 2000, while only 6% contained the inversion before this date. Examination of several smaller inversions (10–30 Mb) failed to reveal a similar phenomenon.

Given access to a complementary pan-genome, we next assessed whether variation in transcription factor binding sites (TFBSs) in the proximal regions of orthologous genes could influence variation in

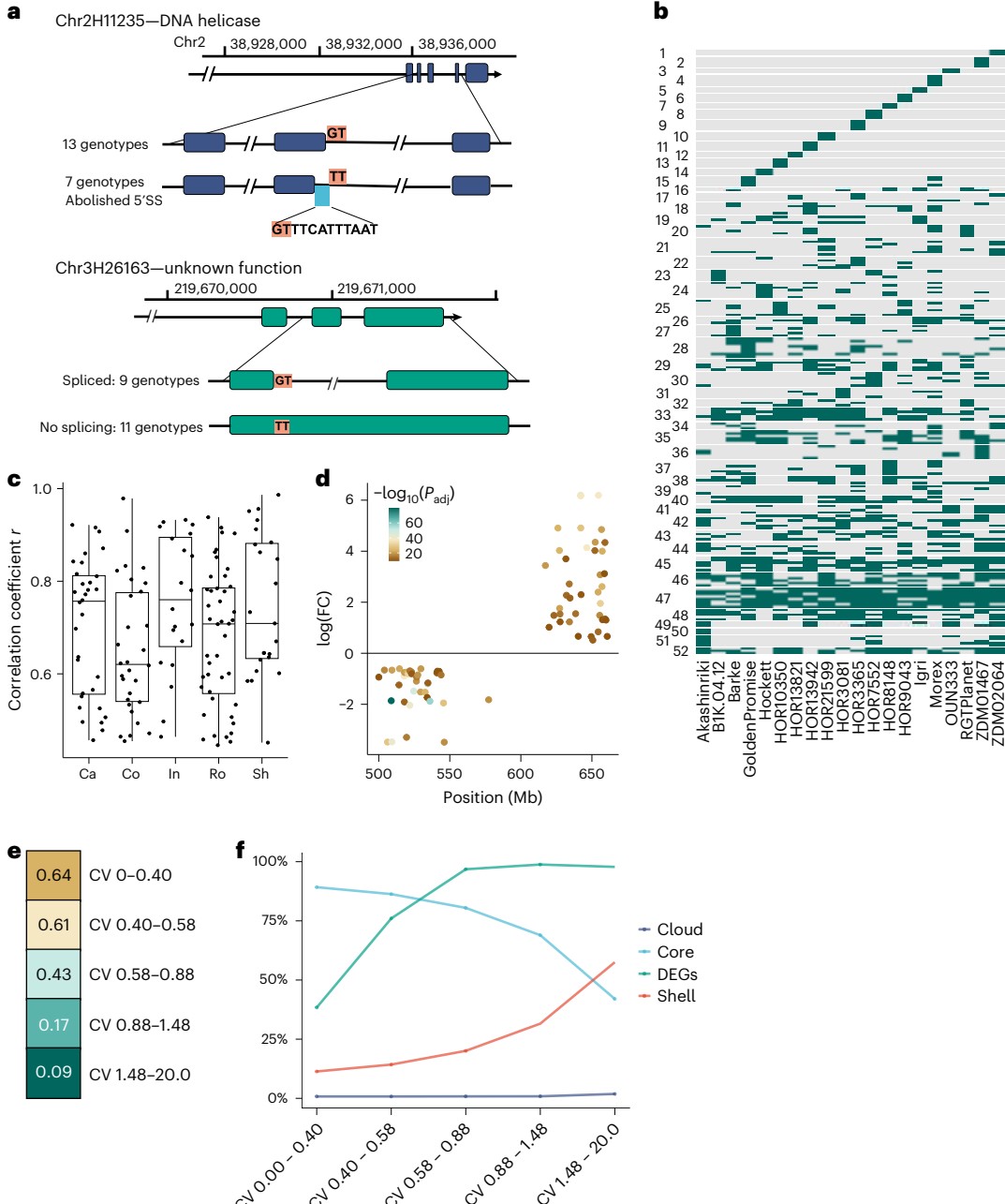

**Fig. 2 | Drivers of transcript abundance variation. a**, Examples of alternative 5′ splice site and intron retention variation among pan-transcriptome genotypes. Top: a 1 nt change (G to T) in seven genotypes caused the selection of an upstream splicing site in chr2H11235. Bottom: splicing was abolished in 11 genotypes in chr3H26163 due to a G to T change. **b**, Heatmap (PAV) of genes with zero transcript abundance in at least one genotype clustered according to similarity. Grey represents zero detected expression. **c**, Correlations between gene copy number and transcript abundance for 98 multiple-copy genes across five tissues. The boxplot whiskers show minimum and maximum values, the upper bound of the box represents the 75th percentile, the lower bound of the box represents the

25th percentile and the centerline represents the median. **d**, Location, magnitude and direction of DEGs across the 141 Mb inversion on chromosome 7H. Statistics were performed using the limma-voom R package with multiple comparison adjustments using the BH procedure. **e**, Heatmap with Pearson correlations between percent identities in TFBSs in upstream 2 kb regions and percent coherence (within ±30%) in expression values computed for sets of ~3,000 genes showing increasing TPM CV. **f**, For each set of ~3,000 genes, the percent content of core, shell, cloud and DEGs is reported. DEGs, differentially expressed genes; SS, splice site.

transcript abundance. We computed the location and occurrence of 30 TFBSs[29–31] in 2,000 bp upstream and 500 bp downstream of the CDS and compared these to a reference set representing all genes of cv. Morex (Methods and Extended Data Fig. 4a). The coherence of expression of orthologous gene pairs was calculated by comparing the percentage of genes whose TPMs are within ±30% of those from cv. Morex to the percent identities of the corresponding TFBSs.

When gene expression is determined mainly by the selected TFBSs, the level of conservation of TFBSs should be highly correlated to transcript abundance. Overall, we observed a Pearson correlation of $r^2 = 0.395$. However, splitting the genes into classes based on their coefficient of variation (CV) of TPM values revealed that genes with low CV (from 0 to 0.4; gene set 1) exhibited a much stronger correlation ($r^2 = 0.635$), while those with high CV (1.48–20; gene set 5) showed a

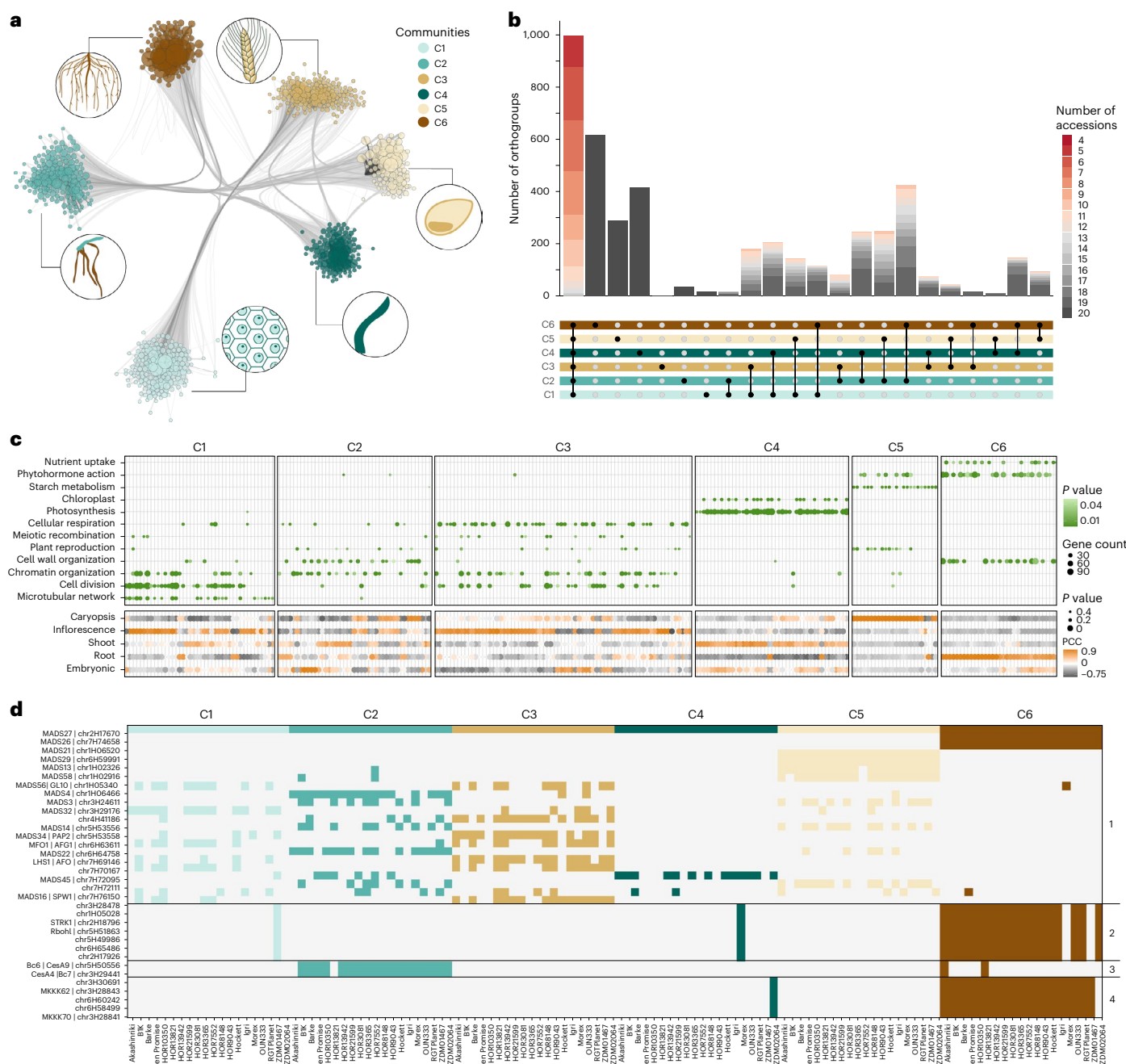

**Fig. 3 | Comparative gene expression and gene network analyses. a**, Illustration of the six main communities identified in the WGCNA analysis. Module–tissue correlation and functional enrichment of the modules included in each of the communities reveal distinct associations with tissues/organs. C1–C3, plant reproductive processes; C4, photosynthesis–shoot; C5, starch metabolism–caryopsis and C6, nutrient uptake–root. **b**, Distribution of 1:1 orthologous groups (exactly 20 genes, 1 from each genotype) across the six communities. Number of accessions refers to the largest cluster within a single community for the respective orthogroup. **c**, Major functional enrichments where the color of the dot represents the FDR-adjusted $P$ value of one-sided Fisher's exact test (top), and the module-tissue PCC where the size of the dot represents the $P$ value of two-sided Fisher's exact test (bottom) for the six main communities. **d**, Accession-specific co-expression distribution of selected barley orthogroups ($y$ axis) associated with distinct functions based on orthology to rice (panel 1, MADS-box protein family; panels 2–4, families discussed in the text) over the six communities and corresponding genotypes ($x$ axis). PCC, Pearson correlation coefficients.

null value ($r^2$ = 0.087; Fig. 2e). The lowest CV fraction is dominated by core genes, while shell, cloud and DEGs are virtually exclusive to the high CV fractions (Fig. 2f). We conclude that for genes with low CV, the similarities in TFBSs in the 2 kb upstream region reasonably account for the coherence of expression. GO enrichment analysis of 2,999 low (0–0.4) and 2,993 high (1.48–20) CV genes showed that the former are enriched for intracellular, housekeeping and essential biological processes, while the high CV class largely comprises terms referring to

defense, response and interaction to environmental cues. We venture that genes showing high tissue specificity and expression variation (high CV) may therefore be influenced more by distal than proximal regulatory elements, as described in mammals[23,24].

## Comparative gene expression and gene network analyses

To investigate expression variation within the single-copy core genes, we conducted a comparative co-expression analysis across all 20

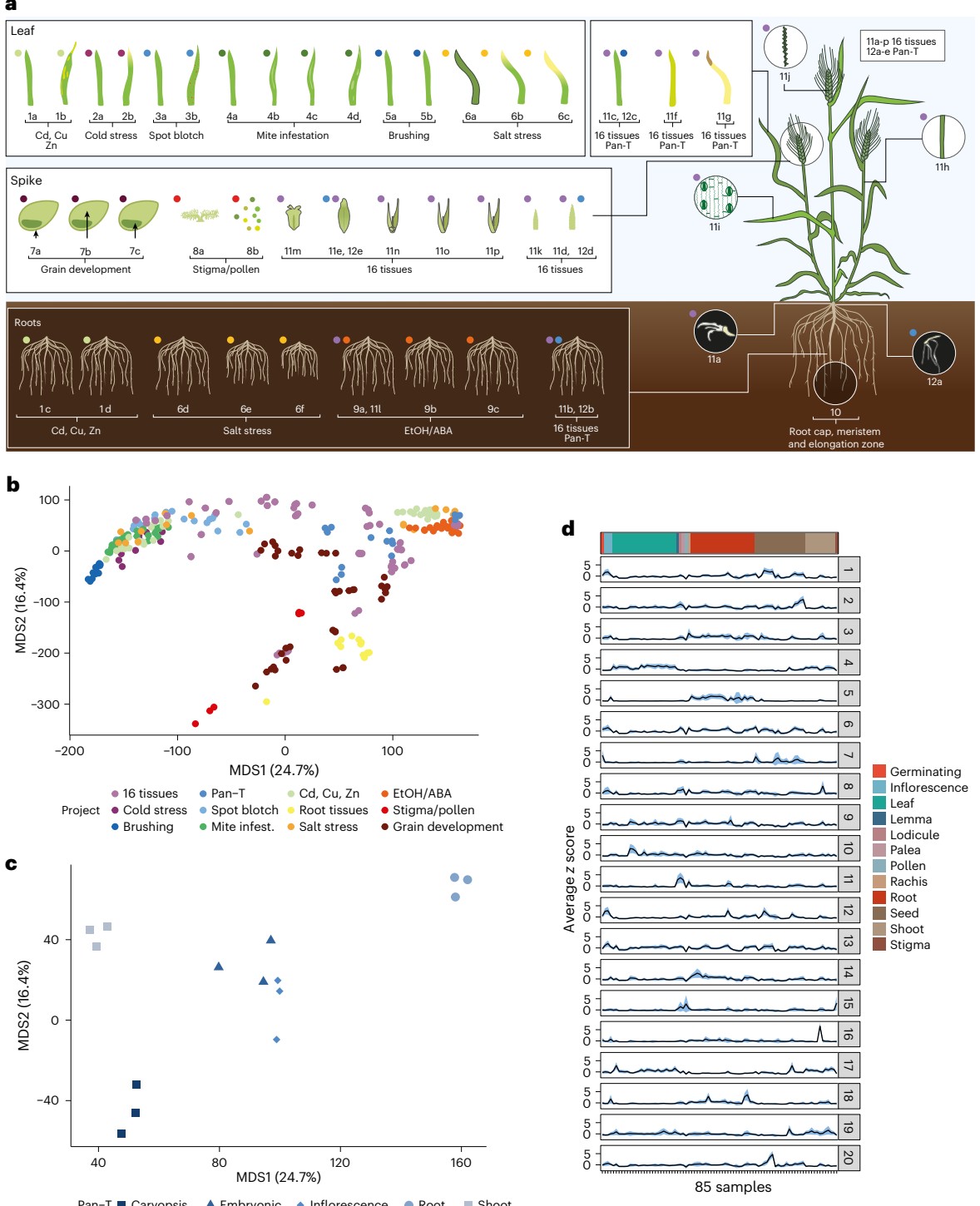

**Fig. 4 | Morex Atlas RNA-seq datasets and variability between samples.**
**a**, Image and description of the samples used to generate the Mx-RTD and used for quantification of differential gene expression. **b**, MDS plot of gene expression data from the replicated RNA-seq datasets and quantified using Mx-RTD. Each of the 12 datasets is represented by the same colored dot in **a** and **b**. The key provides the project identifier associated with each colored dot, and samples are described in full in Supplementary Data 11. **c**, MDS plot of the gene expression data for the five replicated Morex tissues (PRJEB64639) used to develop both GsRTD$_{Morex}$ and PanBaRT20. **d**, Twenty co-expression clusters from 12,292 core:single Morex genes. The black lines indicate average $z$ scores, while the color ribbons show s.e. of the $z$ scores in the clusters, which are ±s.e. of the average. EtOH, ethanol; ABA, abscisic acid.

genotypes. Applying weighted correlation network analysis (WGCNA[32]), we constructed 20 genotype-specific co-expression networks, resulting in a total of 738 modules. Community clustering by the degree of shared orthologs grouped the genotype modules into six major communities, C1–C6 (Methods, Fig. 3a, Extended Data Fig. 5 and community and module assignment in Supplementary Data 9). Modules of three communities (C4, C5 and C6) showed pronounced associations with a single tissue and associated biological processes (Fig. 3c and Extended Data Fig. 6). We found that a notably high number of 12,190 core orthologous groups were partitioned across two or more communities, indicating

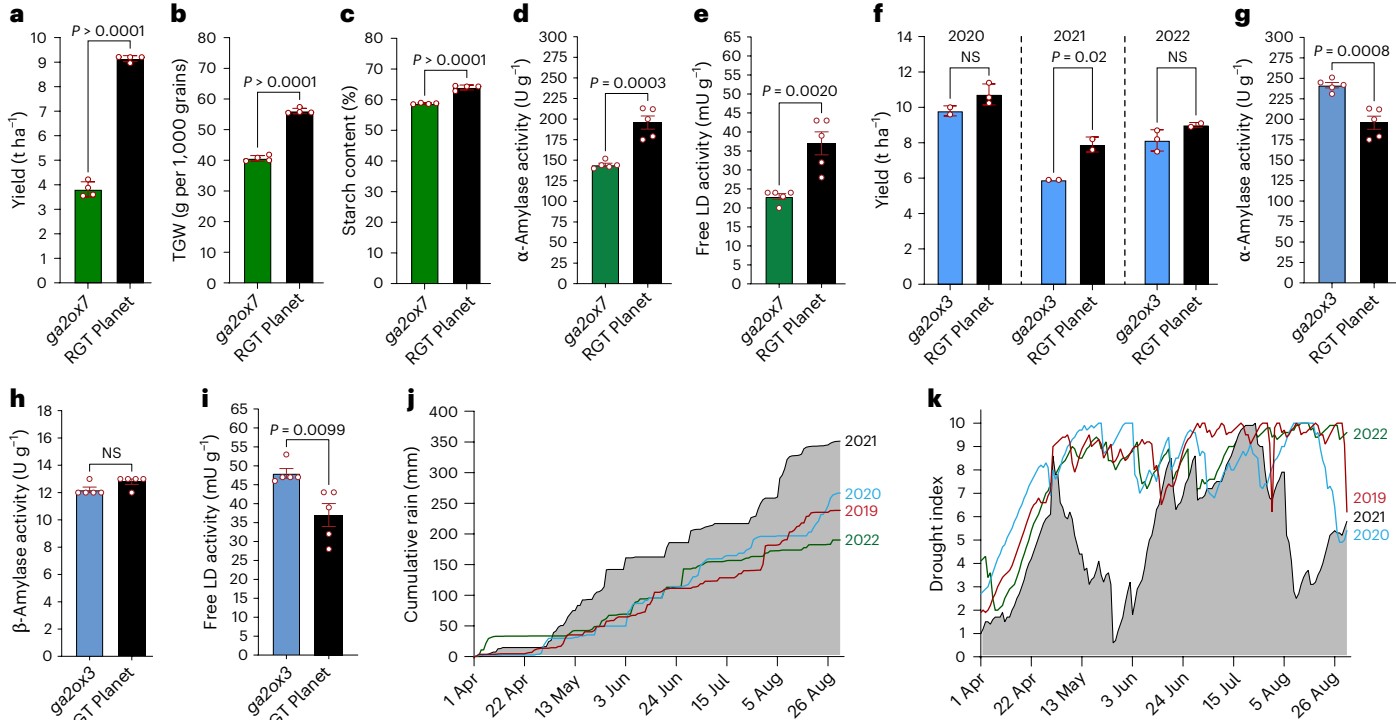

**Fig. 5 | Agronomic and climate data for GA2ox mutants. a–c**, Yield (**a**), TGW (**b**) and starch content (**c**) from field-grown *ga2ox7* mutant (*n* = 4 biological replicates) and RGT Planet control plants (*n* = 4 biological replicates; DK-2019). **d**,**e**, Micromalting data from field-grown *ga2ox7* mutant and RGT Planet control plants (*n* = 5 technical replicates; DK-2022). **d**, α-Amylase activity and **e**, free LD activity. **f**, Yield from *ga2ox3* mutant (2020, *n* = 3; 2021, *n* = 2; 2022, *n* = 2 biological replicates) and RGT Planet control plants (2020, *n* = 2; 2021, *n* = 2; 2022, *n* = 3 biological replicates; DK-2020–2022). **g–i**, Micromalting data from field-grown *ga2ox3* mutant and RGT Planet control plants (DK-2022; *n* = 5 technical replicates). **g**, α-Amylase activity; **h**, β-amylase activity and **i**, free LD activity. **j**, Cumulative rain in millimeters across the years 2019–2022 with the year 2020 in blue. **k**, Drought index (from DMI) across the years 2019–2022 with the year 2020 in blue. Error bars are s.d. All data are provided as mean ± s.d. Two-tailed *t* test was performed to obtain *P* values. LD, limit dextrinase; DMI, Danish Meteorological Institute; NS, not significant.

extensive fine-tuning of biological processes among the 20 genotypes. Supporting this partitioning, orthologues in the same communities showed significantly higher expression correlations (mean Pearson correlation coefficient = 0.895) than those split between two communities (mean Pearson correlation coefficient = 0.702; Welch's *t* test, $P < 1 \times 10^{-16}$). Despite most orthologous groups being split, we observe a clear bias toward conserved expression patterns as illustrated by the nonuniform retention of orthologous genes in larger clusters within a single community (Fig. 3b and Extended Data Fig. 5a). Genes primarily found within a single community were, among others, associated with photosynthesis, nutrient uptake and carbohydrate metabolism (Extended Data Fig. 7).

Split orthogroups were distributed across all major functional categories, biological processes and gene families, including transcription factors (TFs) like MADS-box proteins that integrate many developmental signals (Fig. 3d and Extended Data Fig. 5). Database and literature surveys identified small clusters of functionally related genes that consistently separated into genotype-group-specific patterns[23,24] (Fig. 3d). For example, *Mkkk62* (chr3H28843) and *Mkkk70* (chr3H28841; Fig. 3d) uniquely associate with community 4 in genotype ZDM02064 as opposed to community 6 for all other genotypes. Their orthologs in rice redundantly regulate pollen fertility, leaf angle and cold responses. Similarly, Akashinriki and HOR13821 show divergent community associations for *CesA9* (chr5H50556) and *CesA4* (chr3H29441), orthologs of cellulose synthases in rice that determine cellulose-to-hemicellulose ratio (*Bc6* and *Bc7*; Fig. 3d). A particularly striking example of community divergence comprised 57 orthologous groups specific for the radiation-induced mutant cv. Golden Promise. Of these, 28 were associated with chloroplast development and function in barley or

rice (Supplementary Data 10). Overall, we suggest that both functional diversification and genotype-specific transcriptional responses to our experimental conditions determine the substantial co-expression variation observed in the core orthologs.

**An atlas of gene expression for the genome reference cv. Morex**
Given the defined tissues in PanBaRT20, we explored transcript abundance variation in a more extensive set of tissues and response to treatments from a single genotype, the genome-reference cv. Morex. We leveraged all publicly available (as of September 2023) replicated (≥3) Morex RNA-seq datasets (Supplementary Data 11) derived from various stages of barley development and response to different stimuli (Fig. 4a). We constructed a new RTD from 1,425 Gbp of Morex data and merged this with the GsRTD$_{Morex}$ developed above. We then used this updated Morex RTD (Mx-RTD) to quantify gene and transcript abundance across all 315 Morex samples using Salmon[33]. A multidimensional scaling (MDS) plot again revealed tissues and organs as the major drivers of transcriptional variation. Tissues did respond to applied treatments, but fewer DEGs were involved in the conditional responses (Fig. 4b,c and Extended Data Fig. 8). Co-expression analysis of 12,292 Morex genes that matched the core:single genes in the pan-transcriptome revealed 20 unique clusters exhibiting considerable tissue, organ and conditional specificity, implicating diverse roles in both development and response to treatment (Fig. 4d). Overall, 5,230 Mx-RTD genes had no sequence match in PanBaRT20, likely representing new genes that are characteristic of tissues and conditions not included in the assembly of PanBaRT20. By comparing DEGs ($P \le 0.01$; $\log_2(FC) \ge 1$) from the five cv. Morex pan-transcriptome tissues using both PanBaRT20 and Mx-RTD as reference, we observed a 93.2% overlap

(Extended Data Fig. 9). However, performing the same comparison using specific Morex tissue or treatment datasets not represented in PanBaRT20 revealed fewer significant DEGs when mapping against PanBaRT20. Thus, the heavy metal dataset (PRJNA382490) revealed a 74.7% overlap, pollen and stigma (PRJNA910827) revealed a 54.8% overlap and developing grain samples (PRJNA975859; related to caryopsis in the pan-transcriptome) revealed a 91% overlap. PanBaRT20's major strength is thus in reporting transcript abundance variation among genotypes and may be used in conjunction with the current Mx-RTD to explore wider tissue/treatment effects. We provide this data as 'Morex-GeneAtlas' in EoRNA[34], which provides intuitive transcript abundance plots linked to gene models and associated metadata.

## Exploring the gibberellin 2-oxidase gene family

To illustrate the value of PanBaRT20 and MorexGeneAtlas, we explored the spatial, temporal and genotype-specific expression of members of the gibberellin 2-oxidase (GA2ox) gene family. Gibberellic acid (GA) metabolism is of central importance in breaking grain dormancy and initiating germination as well as plant cell elongation and hence vegetative growth and grain yield[35]. GA metabolism is controlled through complex biochemical pathways involving many different forms of GA. For example, in germinated barley grains, 16 forms of GA have been identified, of which 4 are biologically active. GA2ox enzymes[36], which catalyze the oxidation of the C(2) carbon atom of different GA forms, are particularly important because they can abort GA biosynthesis before bioactive forms are generated and/or inactivate the bioactive forms themselves. These inactivation processes are central to plant survival because they allow a pause in vegetative growth when a germinated grain or young seedling is confronted with a variety of environmental stresses.

We identified ten GA2ox genes, mostly represented in a single copy (Supplementary Data 12), which is not surprising if one assumes that GA2ox enzymes are specific for the many different forms of GA that they oxidize[36]. Of these, chr3H24821 (*GA2ox7* (ref. 37)) and chr3H29540 (*GA2ox3* (ref. 37)) revealed high transcript abundance and widespread variance in genotype-specific expression (Supplementary Data 12). *GA2ox7* was predominantly expressed in the caryopsis in both PanBaRT20 and MorexGeneAtlas, while *GA2ox3* shows genotype-dependent expression in inflorescence, root, shoot and coleoptile and in vegetative tissues in response to wounding, disease, salt or heat stress or oxygen deprivation through waterlogging. In PanBaRT20, *GA2ox7* revealed low expression in the caryopsis of the wild barley accession B1K-04-12 compared to the domesticated genotypes (~50 TPM in B1K-04-12 versus 200 TPM on average for all genotypes), consistent with anthropogenic intervention in barley selection and breeding. Analysis of the upstream TFBS highlighted two zinc finger-type TFBS (AGCTG and WGATAR) present only in B1K-04-12. This class of TF has been reported to regulate GA expression, mostly acting as repressors. In contrast, *GA2ox3* transcript abundance was highly variable across all genotypes. Its promoter shows a high CV (~1.65) with a very poor relationship between TFBS and TPM, suggesting the observed TFBS variation is consistent with the observed differences in transcript abundance. Both *GA2ox3* and *GA2ox7* are predominantly located within co-expression community 6, strongly associated with phytohormone activity (Fig. 3c).

To assess the potential impacts of each gene on agronomic performance and grain quality, we identified FIND-IT[35] knockout mutants in the cultivar RGT Planet and used these in comparative field trials (Supplementary Data 13). The *ga2ox7* mutant exhibited a reduction in yield (Fig. 5a), 1,000-grain weight (TGW; Fig. 5b) and starch content (Fig. 5c), and, in micromalting, strongly reduced α-amylase (Fig. 5d) and free limit-dextrinase activity (Fig. 5e), reflecting reduced levels of bioactive forms of GA in both developing and germinated grain. The *ga2ox3* mutant had wild-type yields in two seasons, but in 2021, when waterlogging affected crop establishment[36] (Fig. 5j,k), showed a strong reduction in agronomic performance (Fig. 5f), along with

increased hydrolytic activity in the grain (Fig. 5g–i). Thus, overexpression of GA2ox7 would be predicted to enhance yields and grain quality, while GA2ox3 overexpression might improve agronomic performance under challenging climatic conditions. Taken together, the pan-transcriptome data identify the repertoire of expressed GA2oxs while providing clues to the specific functions of individual GA2ox genes that can be readily tested in elite germplasm through current or emerging breeding technologies.

## Discussion

Unlike pan-genomes, few plant studies have comprehensively reported the values of assembling and comparing representative transcriptomic datasets from multiple, diverse tissues and individuals that collectively represent a species diversity space. However, such datasets have considerable value in exploring how species diversity underpins the molecular functions that drive biological outcomes, including growth and development, plant architecture and responses to the environment. Previously, variation in transcript abundance among genotypes of biparental populations and diverse inbred lines from single tissues (often without replication) has been generated and used for genetic analysis by eQTL mapping[21,38–40] or associative transcriptomics[41,42], revealing both extensive transcript abundance variation and associations between *cis*- or *trans*-effects and phenotypic traits[43]. More recently, analysis of de novo assembled public RNA-seq datasets from diverse studies and genotypes in barley revealed that 38.2% of the transcripts present in de novo assemblies were absent in the cv. Morex reference, highlighting the issue of 'single reference bias'[44]. Here we illustrate the benefit of establishing and using both genotype-specific and pan-transcriptome reference datasets to interpret transcriptomic data from species-wide genotypic diversity. We show that GsRTDs are powerful and interdependent companions to a pan-genome[8,9], providing experimental evidence for functional genome annotation, enabling robust investigations of transcriptional dynamics within and between genotypes and erecting a platform for exploring the conservation of functional genes, alleles and gene networks among individuals. Our focus on diverse domesticated barley genotypes from across the species range revealed that the introduction, maintenance and loss of functional diversity generally reflect the genomic variation observed in the corresponding pan-genome. However, our network analyses also indicate that the functional consequences of genomic variation on transcript abundance are both complex and genotype-dependent, with cascading effects observed throughout the system. Thus, in the core transcriptome, resident genes exhibit abundant differential expression both within and between tissues and genotypes, suggesting high functional redundancy and intrinsic resilience. We show that a number of evolutionary strategies have contributed to these layers of transcriptional complexity and are reflected in both conserved patterns of co-expression−tissue correlations associated with distinct biological functions−and frequent functional diversification of orthologous genes. Finally, as done previously for wheat[45], to contextualize and complement the pan-transcriptome and facilitate deeper exploration within the biological research community, we provide a 'gene expression atlas' of the barley genomic reference cv. Morex (https://ics.hutton.ac.uk/morexgeneatlas/index.html) along with PanBaRT20 (https://ics.hutton.ac.uk/panbart20/index.html) in EoRNA and illustrate how these resources can be used to underpin biological investigation. All data are available according to findability, accessibility, interoperability and reusability data principles.

## Online content

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

Wenbin Guo [1,20,28], Miriam Schreiber [1,28], Vanda B. Marosi [2,3,28], Paolo Bagnaresi [4,21,28], Morten Egevang Jørgensen [5,28], Katarzyna B. Braune[5], Ken Chalmers[6], Brett Chapman [7], Viet Dang [7], Christoph Dockter [5], Anne Fiebig [8], Geoffrey B. Fincher[6], Agostino Fricano[4], John Fuller[1], Allison Haaning[9,22], Georg Haberer [2], Axel Himmelbach[8], Murukarthick Jayakodi [8,23,24], Yong Jia [7], Nadia Kamal [2,25], Peter Langridge [6], Chengdao Li [7,10,11], Qiongxian Lu [5], Thomas Lux [2], Martin Mascher [8], Klaus F. X. Mayer [2], Nicola McCallum [1], Linda Milne[1], Gary J. Muehlbauer [9], Martin T. S. Nielsen[5], Sudharsan Padmarasu [8], Pai Rosager Pedas[5,26], Klaus Pillen [12], Curtis Pozniak [13], Magnus W. Rasmussen [5], Kazuhiro Sato [14,15], Thomas Schmutzer [12], Uwe Scholz [8], Danuta Schüler[8], Hana Šimková [16], Birgitte Skadhauge [5], Nils Stein [8,17], Nina W. Thomsen[5], Cynthia Voss[5], Penghao Wang[7], Ronja Wonneberger[1,8,27], Xiao-Qi Zhang[7], Guoping Zhang[18], Luigi Cattivelli [4], Manuel Spannagl [2], Micha Bayer [1] ✉, Craig Simpson [1] ✉, Runxuan Zhang [1] ✉ & Robbie Waugh [1,6,19] ✉

[1]International Barley Hub (IBH)/James Hutton Institute (JHI), Dundee, Scotland. [2]Plant Genome and Systems Biology, Helmholtz Center Munich—German Research Center for Environmental Health (PGSB), Neuherberg, Germany. [3]School of Life Sciences, Technical University of Munich, Freising, Germany. [4]Council for Agriculture Research and Economics (CREA) Research Centre for Genomics and Bioinformatics, Fiorenzuola d'Arda, Italy. [5]Carlsberg Research Laboratory (CRL), Copenhagen, Denmark. [6]School of Agriculture, Food and Wine, University of Adelaide, Waite Campus, Urrbrae, South Australia, Australia. [7]Western Crop Genetics Alliance, Food Futures Institute/School of Agriculture, Murdoch University, Murdoch, Western Australia, Australia. [8]Leibniz Institute of Plant Genetics and Crop Plant Research (IPK) Gatersleben, Seeland, Germany. [9]Department of Agronomy and Plant Genetics, University of Minnesota, St. Paul, MN, USA. [10]College of Agriculture, Yangtze University, Jinzhou, China. [11]Department of Primary Industry and Regional Development Western Australia, South Perth, Western Australia, Australia. [12]Chair of Plant Breeding, Institute of Agricultural and Nutritional Sciences, Martin-Luther-University Halle-Wittenberg, Halle (Saale), Germany. [13]Department of Plant Sciences and Crop Development Centre, University of Saskatchewan (USASK), Saskatoon, Saskatchewan, Canada. [14]Institute of Plant Science and Resources, Okayama University, Kurashiki, Japan. [15]Kazusa DNA Research Institute, Kisarazu, Japan. [16]Institute of Experimental Botany of the Czech Academy of Sciences, Olomouc, Czech Republic. [17]Chair of Crop Plant Genetics, Institute of Agricultural and Nutritional Sciences, Martin-Luther-University Halle-Wittenberg, Halle (Saale), Germany. [18]College of Agriculture & Biotechnology, Zhejiang University, Hangzhou, China. [19]School of Life Sciences, University of Dundee, Dundee, UK. [20]Present address: Higentec Breeding Innovation (ZheJiang) Co., Ltd., Lishui, China. [21]Present address: CREA Research Centre for Olive, Fruit and Citrus Crops, Forlì, Italy. [22]Present address: Minnesota Supercomputing Institute, University of Minnesota, Minneapolis, MN, USA. [23]Present address: Texas A&M AgriLife Research Center at Dallas, Texas A&M University System, Dallas, TX, USA. [24]Present address: Department of Soil & Crop Sciences, Texas A&M University, College Station, TX, USA. [25]Present address: Department of Molecular Life Sciences, Computational Plant Biology, School of Life Sciences, Technical University of Munich, Freising, Germany. [26]Present address: DLF, Roskilde, Denmark. [27]Present address: Department of Plant Breeding, Swedish University of Agricultural Sciences, Uppsala, Sweden. [28]These authors contributed equally: Wenbin Guo, Miriam Schreiber, Vanda B. Marosi, Paolo Bagnaresi, Morten Egevang Jørgensen. ✉e-mail: micha.bayer@hutton.ac.uk; craig.simpson@hutton.ac.uk; runxuan.zhang@hutton.ac.uk; robbie.waugh@hutton.ac.uk

## Methods

### Plant material

The 20 barley pan-genome genotypes[8] were used to prepare a barley pan-transcriptome. These are Akashinriki, Barke, ZDM02064, ZDM01467, B1K-04-12, Golden Promise, Hockett, HOR10350, HOR13821, HOR13942, HOR21599, HOR3081, HOR3365, HOR7552, HOR8148, HOR9043, Igri, Morex, OUN333 and RGT Planet. Five different tissue/organ types, including embryo, mesocotyl and seminal roots (together referred to throughout as embryonic tissue), seedling shoot, seedling root, inflorescence and caryopsis, were collected in three biological replicates, amounting to 300 samples. Tissue samples for each genotype were collected at the same growth stage (rather than time after planting), which varied between genotypes.

Forty-five seeds from each genotype, representing three plants for each tissue and each biological replicate, were germinated on water-soaked filter papers in Petri dishes, covered with foil and left at room temperature. Four to eight days postgermination, embryonic tissue at a similar developmental stage was collected by carefully removing the seed coat, leaving the embryo, mesocotyl and seminal roots. The remainder of the germinated seeds were planted in cereal mix compost in seed trays and grown in a glasshouse at nominally 18 °C−16 h light, 400 µmol m$^{-2}$ s$^{-1}$ and 14 °C−8 h dark, 0 µmol m$^{-2}$ s$^{-1}$. Young shoot and root tissues were collected between 14 and 17 days. Plants were then vernalized for 2 weeks at 4 °C−2 h light, 400 µmol m$^{-2}$ s$^{-1}$ and 4 °C−12 h dark, 0 µmol m$^{-2}$ s$^{-1}$, then transplanted into six-inch pots containing cereal mix compost and transferred to a glasshouse and grown under standard barley growing conditions (nominally 18 °C−16 h light, supplemented when appropriate with 400 µmol m$^{-2}$ s$^{-1}$ light from sodium lamps, and 14 °C−8 h ambient dark, 0 µmol m$^{-2}$ s$^{-1}$). The 1–1.5 cm immature inflorescences (Waddington W6−W7) were dissected from the developing spike, and whole developing caryopses were collected at 15–20 days postanthesis (Fig. 1).

### RNA extraction and sequencing

Total RNA extraction was carried out using the Macherey-Nagel NucleoSpin RNA Plant Mini Kit using the manufacturer's instructions. All extractions included the DNAse step. Seedling root and caryopsis tissues included a plant isolation aid (Invitrogen; equal volume of plant isolation aid per gram of tissue) and centrifugation before RNA isolation. Total RNA was quality checked using a Bioanalyzer 2100 (Agilent) and showed an RNA integrity number >8 except for two samples (HOR10350 caryopsis rep 3 and HOR7552 root rep 2), which were not replaced. Hor8148_In3 was identified as mislabeled, thus also being excluded. Strand-specific dUTP libraries and Illumina paired-end 150 bp sequencing were completed by Novogene (UK) Company.

Single-molecule real-time sequencing (SMRT) was performed at IPK Gatersleben. Equal amounts of total RNA from one biological replicate from each of the five sample tissues were pooled, and this was repeated for each of the remaining barley genotypes to produce 20 pooled samples for long-read sequencing. Samples were size fractioned to ~2 kbp, and 300 ng of total RNA per genotype was used to prepare libraries according to the Iso-Seq Express 2.0 workflow using the SMRT-bell Express Template Prep Kit 2.0 (Pacific Biosciences) following the manufacturer's instructions. Each library was sequenced through an individual SMRT cell (one genotype/SMRT cell) on a PacBio Sequel II.

### Assembly of GsRTDs from RNA-seq and Iso-seq reads

We used RNA-seq short-read and Iso-seq long-read data derived from the five tissues to construct GsRTDs for 20 barley genotypes. Short-read RTDs and long-read RTDs were separately assembled using different approaches and then merged to produce comprehensive GsRTDs.

RNA-seq reads were processed using Fastp (v0.20.1)[46] to remove adapters and filter reads with a quality score of less than 20 and a length of less than 30 bases. STAR (v2.7.8a)[47] was applied to map the trimmed reads to the reference genomes[8]. We applied the two-pass

method and used SJs detected in the first mapping pass to improve sensitivity in the second mapping pass. We used Stringtie (v2.1.5)[48] and Scallop (v0.10.5)[49] to construct the transcript assemblies. We then used RTDmaker (https://github.com/anonconda/RTDmaker) to merge the assemblies across samples and filter out low-quality transcripts, including those with noncanonical SJs, SJs supported by fewer than ten reads in fewer than two samples, fragmentary transcripts whose lengths are less than 70% of the gene length, low-abundance transcripts (less than one TPM in fewer than two samples) and mono-exonic antisense fragment transcripts located on the opposite strand of the gene with a length less than 50% of the gene length.

The raw Iso-seq long-read data were processed into full-length non-concatemer (FLNC) reads using the IsoSeqv3 pipeline v3.4.0 (https://github.com/PacificBiosciences/IsoSeq). We mapped the FLNC reads to the reference genome using Minimap2 (v2.24) with maximum intron size '-G 15,000' (ref. 50) and collapsed the mapped reads using TAMA (24 December 2022)[51] to generate transcript models. As described previously[52], we only retained transcripts with high-confidence (HC) SJs and transcript starts and ends. We then conducted a redundancy evaluation to merge transcripts with small variances of 50 bp at the 5′ and 3′ UTR regions using TAMA merge (-a 50 and −z 50).

Finally, we integrated the short-read assemblies into the long-read assemblies by only including transcripts containing new SJs or contributing to new gene loci, resulting in GsRTDs of all 20 genotypes.

### Construction of pan-transcriptome

We used PSVCP (v1.0.1)[17] with default parameters to construct a linear pan-genome using Morex V3 as the reference. New sequences of the remaining 19 genomes were integrated iteratively (Supplementary Data 2). To construct the barley pan-transcriptome (PanBaRT20), the transcripts from 20 GsRTDs were mapped to the PSVCP pan-genome using Minimap2 (v2.24; ref. 50) with the maximum intron size restricted to 15,000 (ref. 50). The unmapped transcripts and transcripts with secondary and supplementary alignments were eliminated. The mapped transcripts were grouped into representative transcript models using the following approach (Extended Data Fig. 1): (1) multiple-exon transcripts with identical intron combinations were merged, and the furthest position of transcript start site/transcript end site of that gene was considered as the start and end of the representative transcript. (2) Overlapped mono-exon transcripts were merged into a single representative transcript by taking their union set. (3) A set of overlapped transcripts located entirely within the intronic regions of other transcripts was treated as a separate gene model. A gene and transcript ID look-up table between PanBaRT20 and GsRTDs was generated based on the transcript grouping processes.

The core genes of PanBaRT20 were defined as those genes represented in all 20 genotypes, the shell genes as those represented in 2–19 genotypes and the cloud genes denoted genes that were unique to a single genotype. These categories were further distinguished based on whether a PanBaRT20 gene matched single or multiple genomic loci within a single genotype. We used Transuite (v0.2.2)[53] to identify the longest ORFs and derived protein sequences. We used InterProScan (v5.59-91.0)[54] to gain insight into the transcript functions from multiple protein databases.

### Transcript quantification and expression analysis

We conducted transcript quantifications using Salmon[33] on 327 RNA-seq samples using PanBaRT20, 20 GsRTDs and BaRTv2 (ref. 16; please note this included RNA-seq data from two genotypes (30 samples) not reported here). We used the 3D RNA-seq (v2.0.1)[55] pipeline to perform differential expression analysis to identify the genes with significant expression changes among five tissues across 20 genotypes. In each tissue, Morex samples were treated as the reference, and all the remaining genotypes were compared to Morex. $F$ test was used to generate a $P$ value across 19 contrast groups. To be considered significant,

a gene must have a Benjamini–Hochberg (BH)-adjusted $P$ value < 0.01 and an absolute $\log_2(FC) \geq 1$ in at least one of the contrast groups.

## Evaluation of expression quantification using PanBaRT20, GsRTDs and GsRTD_Morex

We simulated RNA-seq data (150 bp paired-end reads) to assess the performance and accuracy of expression quantification using different transcriptomes. Ground truth was set during the simulation using Polyester (v1.29.1)[56] as the transcript read counts obtained through GsRTD quantifications of five genotypes (Akashinriki, Barke, Golden Promise, Morex and OUN333). Quantification was carried out based on the simulated reads using PanBaRT20 and the corresponding GsRTDs and GsRTD_Morex. Error assessment was based on relative percentage difference $RPD = 2 \times \frac{|x-x_0|}{|x+x_0|}$ to measure the errors between quantification $x$ and ground truth $x_0$. We used Pearson correlation to evaluate the consistency of expression changes across samples between quantification of the RTDs and ground truth.

## Identification of tandem duplicated genes

In total, 79,580 PanBaRT20 gene sequences were clustered using CD-HIT (v4.8.1) with the following parameter settings: -c 0.95 -r 0 -g 1 -s 0.5. Initial clusters with gene number ≥2 were further filtered as follows: (1) removing clusters with overlapping genes, (2) removing clusters spanning more than 5 Mbp and (3) excluding clusters if any gene copy is from another chromosome. All filtering was done in R.

## Association between CNV and gene expression

The total PanBaRT20 gene number for all 20 accessions was collected from the GsRTD and PanBaRT20 match table (Supplementary Data 15). Clusters with a median expression below ten TPM were excluded, and for the remaining clusters, the Pearson correlation coefficient was calculated on a per-tissue basis. The CDS for CBF2 (chr5H52041) and CBF4 (chr5H52037) was used to confirm the CNV in the genomes and GsRTDs with BLAST 2.15.0 (ref. 57), setting minimum percent identity and coverage to 95% (Supplementary Data 6).

## Inversion analysis

In total, 20 different inversions over 10 Mb in size in relation to the cultivar Morex[8] were aligned on a per-chromosome basis using minimap2 (-2 -I 6G -K 5G -f 0.005 -x asm5 -c --eqx) followed by variant identification using SyRI (v1.6.3)[58] and plotting using plotsr (v0.5.4)[59] with default parameters. Differential gene expression analyses using RNA-seq reads from PRJEB49069 (ref. 21) were as described above. PCR primers as described[8] were used to confirm genotypes with the 141 Mb 7H inversion (Supplementary Data 7). Contrast groups were set up that contained either genotypes with the inversion or without it. Genes with an adjusted BH $P$ value of below $10^{-10}$ and a log FC above 0.5 were considered significant.

## TFBS analysis

Using the barley pan-genome[8], we first selected sets of HC genes consisting of CDS of high quality (Automatic assignment of Human Readable Descriptions score = 3; see https://github.com/groupschoof/AHRD) without potential association to transposable elements (TE; TE-related = 0) and further filtered for the absence of any 'N' bases by scanning each CDS. This selection resulted in 22,000–23,000 filtered CDS for each accession.

We defined a set of genes in Morex V2 that had a bona fide counterpart in Morex V3 by BLAST Best Reciprocal Hits, setting the minimum percent identity and coverage to 95% and the expected value to $10^{-40}$ using the high-quality 22,541 Morex_V2 proteins as the query and 70,355 Morex_V3 proteins (including isoforms) as the database. Close correspondence between Morex_V2 and V3 was found for 15,001 genes, and whenever necessary, these were used to trace the counterparts of Morex V3 genes in the pan-genome V1 genotypes.

For each of the 15,001 V2_V3 corresponding genes, we conducted a BLAST Best Reciprocal Hits analysis of Morex versus each genotype filtering for CDS identity ≥99% and no alignment gaps (100% coverage for query and participants). This defined 19 sets of close orthologous pairs ranging from 12,071 to 10,330 pairs. Then, for each pairwise comparison, 2 kb upstream sequences from CDS start (based on coordinates as available in general feature format (gffs)) were retrieved and split into four 500 bp bins. Furthermore, a 500 bp downstream region was also captured and compared to prove substantial identity in the sequences of the selected genes.

Thirty TFBS consensus sequences (cis-elements, in direct orientation) have been selected based on recent data[29] and further literature[30,31] to account for the major plant TF classes of the most well-known cis-elements. The following TFBS consensus sequences were considered: (1) AP2-DREB-RCCGAC, (2) AP2-ERF−CGCCGCC, (3) AP2-TOE2−MTCGTA, (4) B3-AuxRE2−TGTCGG, (5) B3-AuxRE−TGTCTC, (6) B3-LEC-AB3−CATGCA, (7) B3-RAV−CACCTG, (8) bZIP-A-box−TACGTA, (9) bZIP-C-box−GACGTC, (10) bZIP-G-box−CACGTG, (11) bZIP-GCN4−TGASTCA, (12) C2C2-ZF-DOF−AAAGY, (13) C2C2-ZF-GATA−WGATAR, (14) C2C2-ZF-YABBY−WATNATW, (15) C2H2-ZF−CAGCT, (16) GARE−TAACARA, (17) HD-WOX13−CAATCA, (18) HD-ZIP−AATNATT, (19) LOB−TCCGGA, (20) MRECHS−AACCTA, (21) MYB-GARP-B-ARR−AGATWCG, (22) MYB-R2R3-MBSI−CNGTTR, (23) MYB-R2R3-MBSIIG−GKTWGGTR, (24) MYB-R2R3-MBSII−GKTWGTTR, (25) MYB-related−AGATAT, (26) SPL−CGTAC, (27) STY1−CCTAGG, (28) TCP−GGNCCC, (29) trihelix−GRWAAW and (30) WRKY−TTGACY. This list corresponds to the positions from 1 to 30 reported on the $x$ axis in Extended Data Fig. 4a.

Degenerate International Union of Pure and Applied Chemistry bases were set according to the appropriate position weight matrix to account for ambiguities. By scanning for occurrences of the 30 TFBSs in the four 500 bp bins, we obtained (via SeqPattern (v1.32) and OmicsBox (v3.1.2)) overall statistics by comparing the location and occurrences in all pairwise comparisons among Morex and the other 19 genotypes. These data were used to evaluate the expression coherence for ortholog pairs defined as the percent of genes whose TPMs are in the range of plus or minus 30% relative to their Morex orthologs. 'Expression coherence' was computed on these orthologous gene pairs.

The entire TPM dataset was used to compute the CV, resulting in minimum, median, mean and maximum values of 0.16, 0.71, 0.99 and 15.07, respectively, splitting the genes into five classes based on CV ranges (0–0.4, 0.4–0.58, 0.58–0.88 and 0.88–20 with the first and last values set to 0 and 20, respectively) and close to 3,000 members in each class.

## Network analysis

Of the 13,680 single-copy genes, 13,652 genes passed a filter for lowly expressed genes (TPM >0.5 in minimum of two biological replicates) in all accessions. From the original 297 samples, PCA revealed an outlier (ZDM01467_In2) that was removed from further analysis. R package DESeq2 (v1.34.0)[60] was used to perform variance-stabilizing transformation for normalization, applying a design '-Tissue' with the argument 'blind = FALSE'. Co-expression networks were built for all accessions' single-copy orthologs separately using the WGCNA package (v1.69). The soft power threshold used for scale-free topology was determined automatically using 'pickSoftThreshold'. Unsigned networks were calculated with Pearson correlation to obtain the adjacency matrix that accounted for both positive and negative correlations among gene pairs by applying methods dynamicTreeCut and TOMtype with a minimum module size = 30, 'unsigned' network type and otherwise default settings. This resulted in 20 networks, with different module numbers for each accession. This approach was followed by merging closely clustered modules by a cut height of 0.1, which resulted in the final set of modules for

each accession. A total of 738 modules across all accessions were identified. In total, 91 genes with insufficient variation have been filtered out from each of the genotype-specific networks using the 'goodSamplesGenes' function, leaving 13,561 single-copy genes for the following analysis.

Module eigengenes were calculated using the 'moduleEigengenes'. Pearson correlation with Fisher's exact significance test was calculated among a binarized tissue table and the eigengenes (Fig. 3c and Extended Data Fig. 5). Mercator4 (v5.0)[61] functional annotation was performed, and enrichment with over-representation analysis (ORA) of sets of module-genes was done using the R package clusterProfiler (v4.6)[62] with BH false discovery rate (FDR) correction $P$ value cutoff 0.05 (Fig. 3c and Extended Data Figs. 5 and 6). ORA results are shown only for those modules where module size enabled the enrichment analysis (Extended Data Figs. 5 and 6).

To compare the shared number of orthologs across accession-specific network modules, cosine similarity was calculated for each module pair (Extended Data Fig. 6) using Python (v3.10.12). Across all accessions, a meta-network was established by computing edges using the cosine similarity values and nodes as modules using NetworkX (v3.1)[63] and clustered further into six distinct communities via Louvain community detection[64]. The final network layout was calculated using the Netgraph (v4.12.11)[65] package with the edge bundle option (Fig. 3a). Ortholog detection among *Oryza sativa*[66], *Arabidopsis thaliana*[67] and selected barley accessions Morex, Akashinriki and B1K representative protein sequences was performed using OrthoFinder (v2.5.5)[68] with default parameters.

### Creation and analysis of barley 'MorexGeneAtlas'

All public cv. Morex datasets with ≥3 replicates were retrieved from the European Nucleotide Archive (ENA). Discovery of datasets using the Representational State Transfer Application Programming Interface is available at https://www.ebi.ac.uk/ena/portal/api/search. Samples came from the following study accessions: PRJEB29972, PRJEB52944, PRJNA315041, PRJNA326683, PRJNA382490, PRJNA639318, PRJNA661163, PRJEB14349, PRJNA910827, PRJNA975859 and PRJNA589222 (Supplementary Data 11). We also included the 15 samples from the Morex pan-transcriptome dataset listed under the study accession PRJEB64639. Raw RNA-seq reads were trimmed using fastp (v0.20.0)[46] and mapped to the Barley cv. Morex V3 (2021) genome[69] using STAR (v2.7.8a)[47]. Transcripts were assembled using Stringtie (v2.1.5)[48] and Scallop (v0.10.5)[49]. The resulting annotation was filtered to remove redundant and fragmented transcripts, transcripts with poorly supported SJs and low-expressed transcripts. The resulting assemblies were merged and filtered using the RTDmaker (v0.1.5; https://github.com/anonconda/RTDmaker), forming Mx-RTD. Quantification of expression against both Mx-RTD and PanBaRT20 was carried out using Salmon (v0.14.1) in quasi-mapping mode[33].

The Morex Gene Atlas and database were prepared using the LAMP configuration (Linux, Apache, mySQL and Perl) and created as described previously[34]. Significant differential gene expression and differential alternative splicing analysis were performed using the 3D RNA-seq App (https://3drnaseq.hutton.ac.uk/app_direct/3DRNAseq/)[55]. All scripts used in the construction and analysis of the Morex Atlas have been made available at https://github.com/cropgeeks/barleyPantranscriptome.

### GA metabolic pathway analysis

Gene sequences of the GA metabolic pathway genes (based on IBSC2017 cv. Morex assembly) as reported previously[36] were used to BLAST against the genome assemblies of the 20 barley accessions used in the updated pan-transcriptome study[9]. Identified gene sequences were BLASTed against PanBaRT20, and individual gene expression values were extracted.

### FIND-IT screening

Barley FIND-IT variants *ga2ox3*[w106*] (ID CB-FINDit-Hv-034) and *ga2ox7*[w107*] (ID CB-FINDit-Hv-035) were identified and isolated in the RGT Planet mutant library as described[70].

### Barley field trials and yield performance

FIND-IT variants and controls were grown in field trial plots for agronomic evaluation (7.5 m$^2$ plots) in Denmark. Grain from field plots was harvested and threshed using a Wintersteiger Elite Plot Combine (2019 and 2020) and a Wintersteiger Classic Plus Combine (2021 and 2022; Wintersteiger AG), and grains were sorted by size (threshold, 2.5 mm) using a Pfeuffer SLN3 sample cleaner (Pfeuffer GmbH).

### Agronomic performance evaluation

Mature dry grains from field-grown plants were analyzed to determine TGW using a MARVIN digital seed analyzer (GTA Sensorik GmbH), and grain quality (starch, protein and water content) was measured using near infrared technology (Foss Infratec 1241 analyzer).

### Weather data

Weather data for the field trial region of the respective years were extracted from the weather archive of the Danish Meteorological Institute (https://www.dmi.dk/vejrarkiv/).

### Micromalting and hydrolytic enzyme activity analysis

Nondormant barley samples of RGT Planet and the FIND-IT variants *ga2ox3*[w106*] and *ga2ox7*[w107*] were placed in individual containers, each holding 100 g of seeds, and submitted into 16 °C fresh water to reach 33% moisture content on day 1 and 43% moisture content on day 2. The actual water uptake of individual samples was determined as the weight difference between initial water content, measured with the Foss 1241 NIT instrument (Foss A/S), and the sample weight after surface water removal. Following the last step, the barley samples were maintained at a degree of steeping of 43% for 4 days. After each 24 h, samples were checked for moisture content and sprayed with additional water to overcome possible respiration loss. After the germination process, the barley samples were kiln dried in a Curio kiln using a two-step ramping profile. The first ramping step started at a set point of 27 °C and a linear ramping at 2 °C h$^{-1}$ to the breakpoint at 55 °C using 100% fresh air. The second linear ramping was at 4 °C h$^{-1}$ reaching a maximum at 85 °C. This temperature was kept constant for 90 min using 50% fresh air. The kiln dried samples were deculmed using a manual root removal system from Wissenschaftliche Station für Brauerei. Before enzyme activity analysis, the germinated grain samples were milled using a standard IKA Tube Mill 100. All measurements of enzyme activity in germinated barley grains were made within 48 h after milling of the sample. The enzymatic activity of α-amylase, β-amylase and free limit dextrinase was measured using the Megazyme 'Ceralpha', 'Betamyl-3' and 'PullG6' methods, respectively, that were modified for the Gallery Plus Enzyme Master (Thermo Fisher Scientific).

### Gene and transcript naming

Multiple historical genomic and transcriptomic experiments have led to a range of barley gene name annotations for the same gene. Morex-GeneAtlas provides a barley gene name 'look-up' function (https://ics.hutton.ac.uk/morexgeneatlas/relator.html). Submitting a legacy barley gene identification name returns a table of matching transcript identification names from MLOCs, JLOCs, HORVUv1, HORVUv3, BARTv1 and BARTv2 datasets. It also provides transcript identification names from the GsRTD$_{Morex}$, HvMxRTD and PanBaRT20 datasets from the current paper.

### Reporting summary

Further information on research design is available in the Nature Portfolio Reporting Summary linked to this article.

## Data availability

All raw data are available through the ENA (https://www.ebi.ac.uk/ena/browser/home). Raw Illumina reads are available as BioProject accession PRJEB64639; Iso-seq CCS reads are available as BioProject accession PRJEB64637. All data underpinning PanBaRT20 and Morex-GeneAtlas are also available in EoRNA (https://ics.hutton.ac.uk/panbart20/index.html) and https://ics.hutton.ac.uk/morexgeneatlas/index.html. The Morex V3 pseudomolecules are available at https://doi.org/10.5447/ipk/2021/3. Supplementary Data 1–15 have been uploaded to figshare repositories and can be accessed at https://doi.org/10.6084/m9.figshare.28035638 (ref. 74).

## Code availability

Scripts used for data analysis have been made available at https://github.com/cropgeeks/barleyPantranscriptome, https://github.com/vanda-marosi/PanBarleyNetworks and https://github.com/WCGA-Murdoch/Barley-phenology-2023. Corresponding DOIs have been minted with Zenodo (https://doi.org/10.5281/zenodo.13961253 (ref. 71), https://doi.org/10.5281/zenodo.13961795 (ref. 72) and https://doi.org/10.5281/zenodo.13950149 (ref. 73)).

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

## Acknowledgements

We specifically acknowledge S. Horne at the James Hutton Institute (JHI) for original artwork. We acknowledge the following organizations for financial support: JHI and University of Dundee—the Scottish Government RESAS project KJHI-B1-2, BBSRC project (BB/X018636/1, BB/S020160/1, ERA-CAPS BB/S004610/1 and BB/S020160/1); Helmholtz Center Munich—BMBF project (031B0884B and 031B0190); CREA—PRIMA project GENDIBAR; Adelaide and Murdoch Universities—GRDC project (UMU1806-002RTX); Carlsberg Research Laboratory—Carlsberg Foundation project (CF15-0236, CF15-0476 and CF15-0672); IPK Gatersleben—DFG project 'BARN' STE1102/15-2, BMBF project (031B0190A and 031B0884A); University of Minnesota—NSF ERA-CAPS project (1844331); Martin-Luther-University Halle-Wittenberg—DFG project (433162815); University of Saskatchewan—Genome Canada project (CTAG2); Okayama University and Kazusa DRI—JSPS project (23H00333); Institute of Experimental Botany of the Czech Academy of Sciences—ERDF project (CZ.02.1.01/0.0/0.0/16_019/0000827); and Zhejiang University—CNSF project (31620103912). We acknowledge the Research/Scientific Computing teams at the JHI and National Institute of Agricultural Botany for providing computational resources and technical support for the 'UK's Crop Diversity Bioinformatics HPC' (BBSRC grant BB/S019669/1), the use of which has contributed to the results reported within this paper.

## Author contributions

M.B., L.C., C.D., C.L., P.L., M.M., K.F.X.M., G.J.M., C.P., K.P., K.S., H.S., C.S., M. Spannagl, N.S., P.W., R. Waugh, G.Z. and R.Z. conceived the study. C.S., G.J.M., N.S., R. Waugh and R.Z. designed the study. A. Haaning, J.F., N.M., M. Schreiber and R. Wonneberger grew and sampled the biological materials and generated RNA. A. Himmelbach, K.C., L.C., C.D., C.L., P.L., G.J.M., K.P., P.R.P., C.P., S.P., T.S., K.S., B.S., N.S., C.S., R. Waugh, G.Z., R.Z. and X.-Q.Z. provided RNA-seq and Iso-seq data. M.B., G.J.M., C.S., N.S. and R. Waugh coordinated the experiments and RNA-seq. N.S. and M.M. provided early access to the barley Pan Genome V2.0 data to assist analysis. M.B., G.H., W.G., M.J., M.M., K.F.X.M., M. Spannagl, M. Schreiber and R.Z. supervised and conducted sequence preprocessing, quality control and downstream analyses. W.G. and R.Z. developed and evaluated the reference transcript datasets and generated all transcript quantifications and comparisons leading to the definition of core, cloud and shell transcripts. M.B., W.G. and M. Schreiber developed the linear pan-genome framework used for transcript clustering. M.B., W.G., G.H., T.L., M. Spannagl and M. Schreiber supervised and conducted GO analysis and transcript annotation. B.C., V.D., K.B.B., Y.J., M.E.J., Q.L. and M. Schreiber conducted the CNV analysis. A. Haaning, M. Schreiber, H.S. and R. Wonneberger conducted the expression and Hi-C analysis of the 7H inversion. A. Fricano, L.C. and P.B. performed the promoter analyses. V.B.M., K.F.X.M., N.K. and M. Spannagl supervised and performed the WGCNA and network analyses. M.B., L.M. and C.S. downloaded and reanalyzed public Morex RNA-seq data and developed the EoRNA database. M.E.J., C.D., M.T.S.N., C.V., M.W.R., K.B.B., G.B.F. and N.W.T. developed the biological story around GA2ox3 and GA2ox7. M.B., A. Fiebig, W.G., U.S. and D.S. supervised data management and submission. P.B., W.G., V.B.M., M. Schreiber, C.S., R. Waugh and R.Z. wrote the paper. R. Waugh and R.Z. coordinated the consortium. All authors read and commented on the paper.

## Competing interests

K.B.B., C.D., G.B.F., M.E.J., Q.L., M.T.S.N., P.R.P., M.W.R., B.S., N.W.T. and C.V. are employees of the Carlsberg Research Laboratory. The other authors declare no competing interests.

## Additional information

**Extended data** is available for this paper at https://doi.org/10.1038/s41588-024-02069-y.

**Correspondence and requests for materials** should be addressed to Micha Bayer, Craig Simpson, Runxuan Zhang or Robbie Waugh.

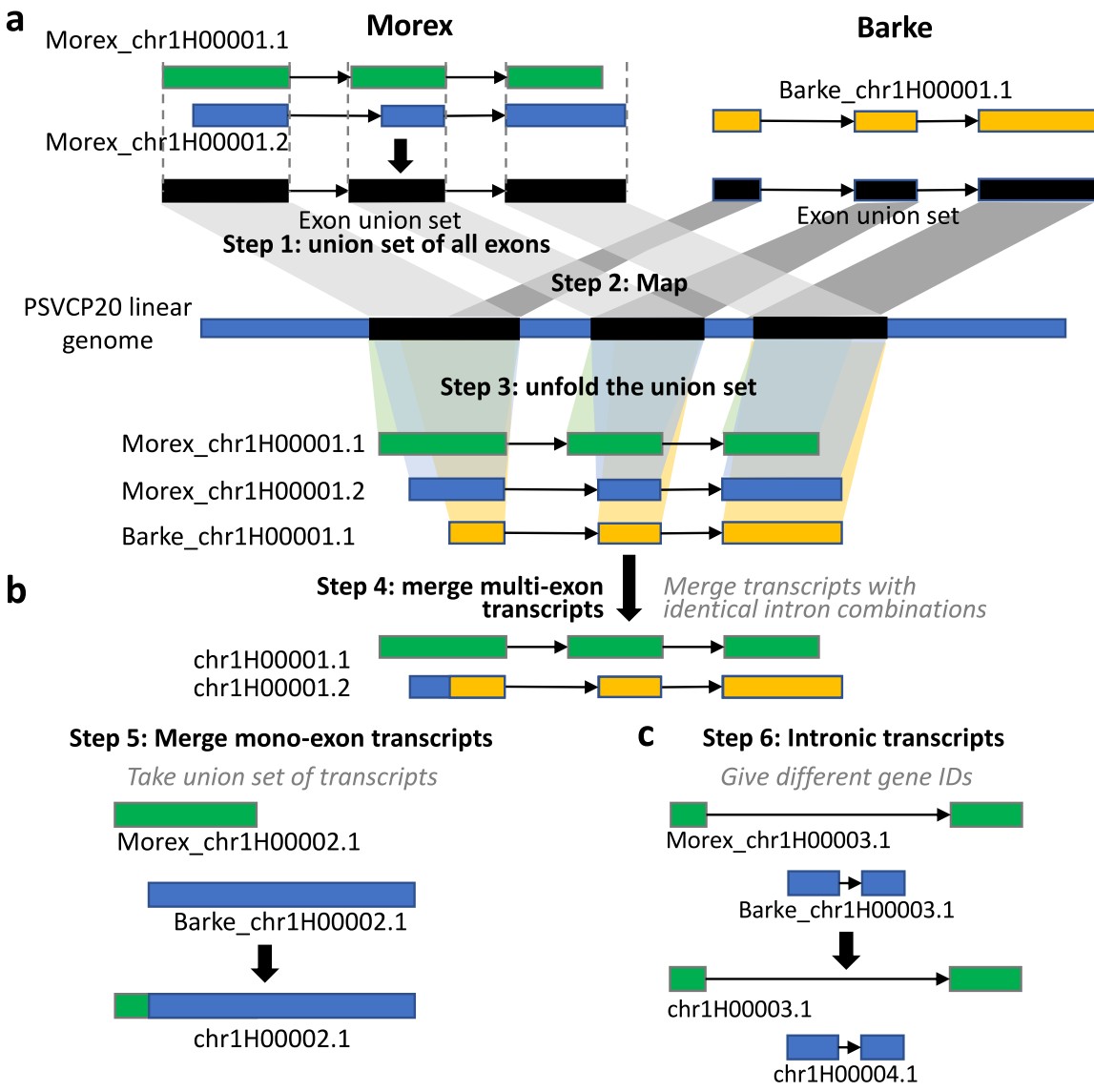

**a**

Morex_chr1H00001.1

**Morex**

**Barke**

Barke_chr1H00001.1

Morex_chr1H00001.2

Exon union set

Exon union set

**Step 1: union set of all exons**

**Step 2: Map**

PSVCP20 linear genome

**Step 3: unfold the union set**

Morex_chr1H00001.1

Morex_chr1H00001.2

Barke_chr1H00001.1

**b**

**Step 4: merge multi-exon transcripts**

*Merge transcripts with identical intron combinations*

chr1H00001.1
chr1H00001.2

**Step 5: Merge mono-exon transcripts**

*Take union set of transcripts*

Morex_chr1H00002.1

Barke_chr1H00002.1

chr1H00002.1

**c**

**Step 6: Intronic transcripts**

*Give different gene IDs*

Morex_chr1H00003.1

Barke_chr1H00003.1

chr1H00003.1

chr1H00004.1

**c**

**Look-up table**

| PanBaRT20_gene_id | PanBaRT20_transcript_id | GsRTD_gene_id | GsRTD_transcript_id |
|---|---|---|---|
| chr1H00001 | chr1H00001.1 | Morex_chr1H00001 | Morex_chr1H00001.1 |
| chr1H00001 | chr1H00001.2 | Morex_chr1H00001; Barke_chr1H00001 | Morex_chr1H00001.2; Barke_chr1H00001.1 |
| chr1H00002 | chr1H00002.1 | Morex_chr1H00002; Barke_chr1H00002 | Morex_chr1H00002.1; Barke_chr1H00002.1 |
| chr1H00003 | chr1H00003.1 | Morex_chr1H00002 | Morex_chr1H00002.1 |
| chr1H00004 | chr1H00004.1 | Barke_chr1H00003 | Barke_chr1H00003.1 |

- **Core genes:** match 20 genotypes
- **Shell genes:** match 2-19 genotypes
- **Cloud genes:** match 1 genotype

- **Single copy:** only one gene in a genotype matched to a PanBaRT20 gene
- **Multiple copy:** multiple genes in the same genotype matched to a PanBaRT20 gene

**Extended Data Fig. 1 | See next page for caption.**

**Extended Data Fig. 1 | Construction of PanBaRT20.** The Morex and Barke GsRTDs were used as examples to illustrate the construction of PanBaRT20 from 20 GsRTDs. **a**, The transcripts in each GsRTD gene were collapsed into an exon union set (step 1). The union sets of all the GsRTD genes were mapped to the PSVCP pan-genome using Minimap2 (step 2). This ensured that all the transcripts from the same gene were mapped to the same genomic loci on the PSVCP pan-genome (step 3). **b**, The overlapped transcripts were assigned the same gene ID. The multiple-exon transcripts that shared identical intron combinations were merged, and the furthest start and end of these transcripts were taken as the transcript start site (TSS) and end site (TES) of the merged transcript (step 4). The overlapped mono-exon transcripts were merged into one transcript with the furthest starting and ending as the TSS and TES (step 5). If a set of overlapped transcripts were located entirely within the introns of other transcripts, they were assigned a separate gene ID (step 6). **c**, After assigning new gene and transcript IDs to the PanBaRT20 gene models, a look-up table was created to record the gene and transcript associations between PanBaRT20 and 20 GsRTDs.

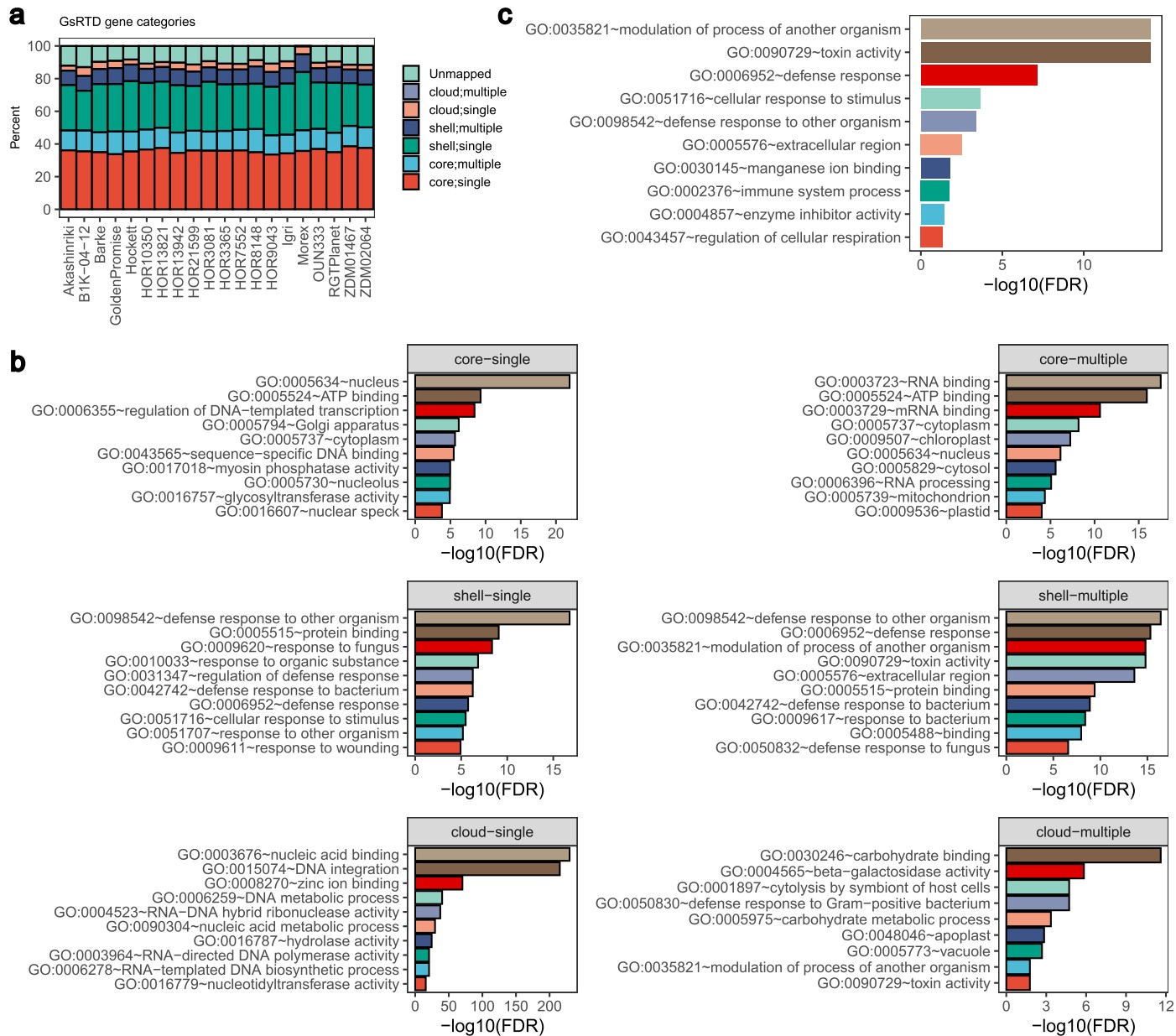

**Extended Data Fig. 2 | Characteristics of GsRTDs and PanBaRT20. a,** The percentages of GsRTD genes in 20 genotypes that matched core-single copy, core-multiple copy, shell-single copy, shell-multiple copy, cloud-single copy and cloud-multiple copy genes in PanBaRT20. **b,** Significant gene ontology (GO) enriched terms of PanBaRT20 genes in core, shell and cloud gene categories. **c,** Significant GO enriched terms of 2,925 PanBaRT20 genes with zero TPM in any tissue in 1–19 genotypes.

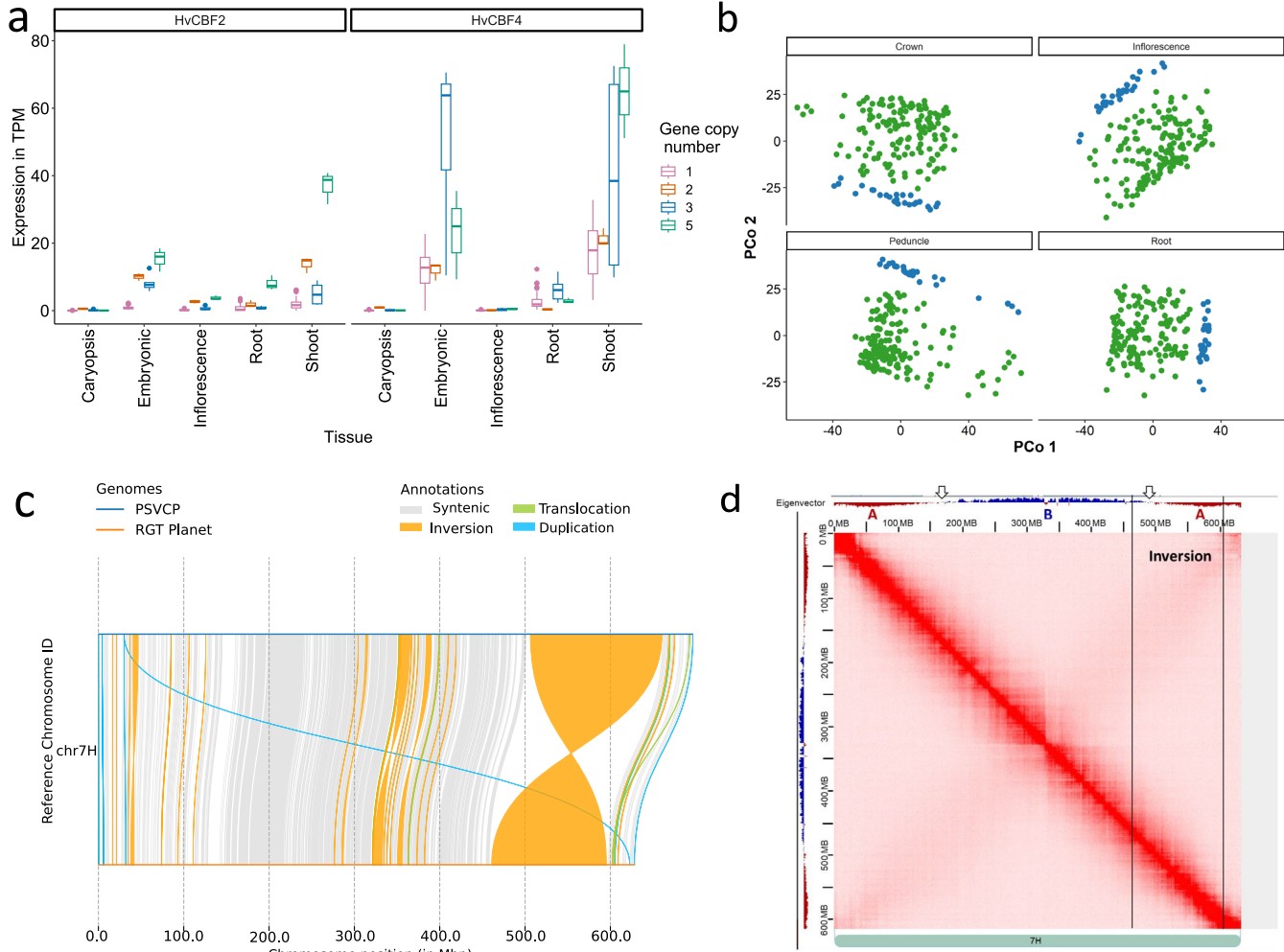

**Extended Data Fig. 3 | Drivers of variation in transcript abundance.**
**a**, Expression of HvCBF2 and HvCBF4 by tissue and copy number. Each genotype had 3 biological replicates per tissue, except for HOR10350 (caryopsis n = 2), HOR7552 (root n = 2) and HOR84148 (inflorescence n = 2). Significant correlation can be found for HvCBF2 in the coleoptile ($r^2$ = 0.85), inflorescence ($r^2$ = 0.62), root ($r^2$ = 0.5) and shoot ($r^2$ = 0.71) and for HvCBF4 in the inflorescence ($r^2$ = 0.44) and shoot ($r^2$ = 0.49). The boxplot whiskers show minimum and maximum values, the upper bound of the box represents 75th percentile, the lower bound 25 percentile and the centerline the median. **b**, Principal component cluster analysis of the genes expressed in the inversion. The region contained 2508 expressed PanBaRT genes in at least one of the four tissues. The PCA splits the 201 genotypes into two clusters in every tissue corresponding to those genotypes carrying the inversion versus those carrying the wild type. **c**, Alignment of chromosome 7H between cultivar RGT Planet and the linear pan-genome generated through PSVCP highlighting the genomic 140 Mb inversion. **d**, Hi-C interaction matrix of chromosome 7H from the cultivar Morex. The division into A and B compartments emerges as principal component 1 in a PCA of Hi-C interaction frequencies. The A/B compartment boundary on the long arm of 7H is quite distinct and part of the inversion in RGT Planet.

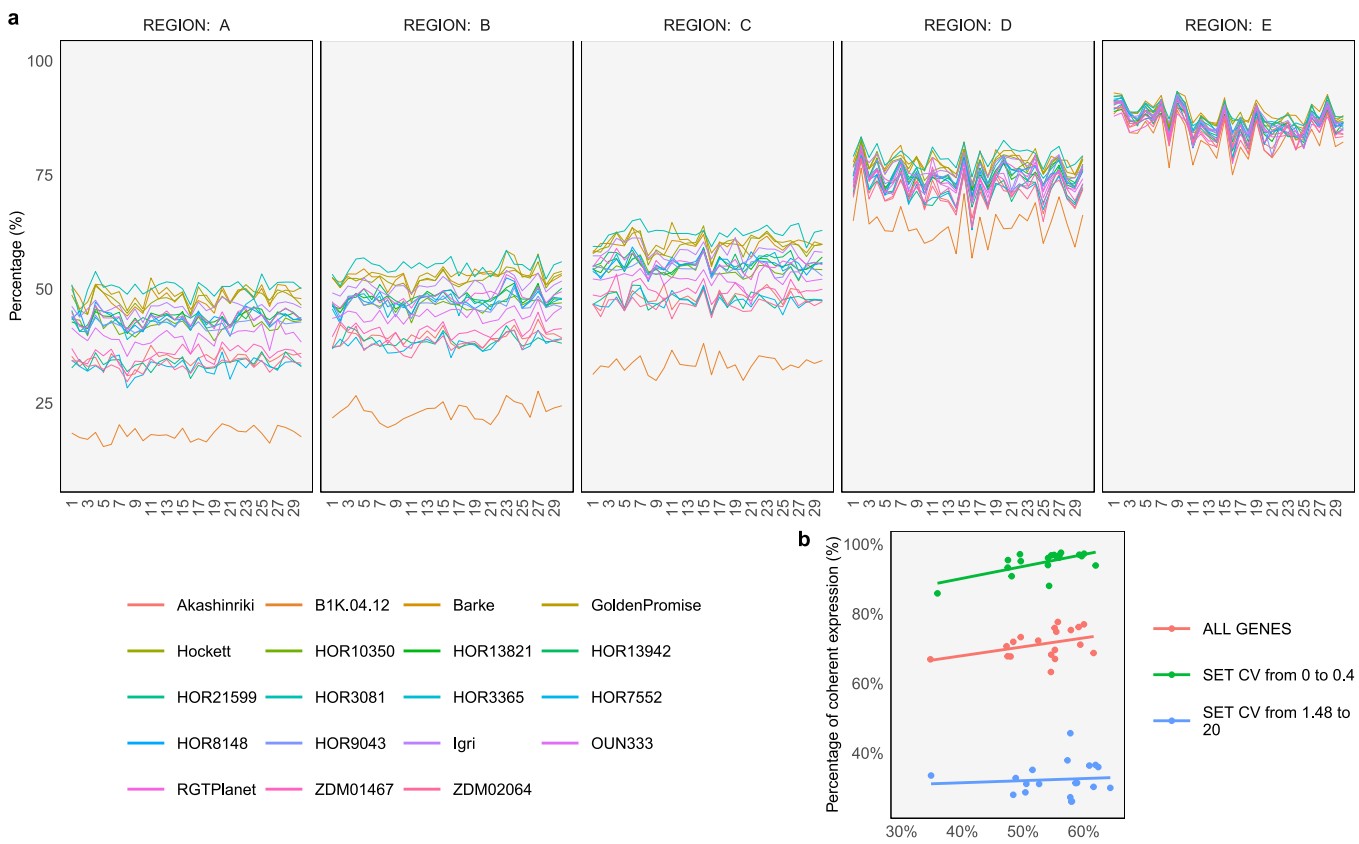

**Extended Data Fig. 4 | Transcription factor binding site analysis. a**, Percent identities for transcription factor binding sites (TFBS) in 500 bp segments of 2 kb upstream (regions from A to D) and 500 bp downstream (region E) of start codon computed for each genotype vs. Morex. The x axis indicates 30 TFBS consensus sequences with a code from 1 to 30 (see Methods for details). **b**, Percent TFBS identities against percent coherent expression in each pairwise comparison (individual genotypes vs. Morex, each pair represented by a dot) for all genes (15,001 genes Pearson correlation value = 0.395), genes with low coefficient of variation (CV; 0–0.4 CV, 2,999 genes, corr. = 0.695) and genes with high CV (1.48–20 CV, 2,994 genes, corr. = 0.087).

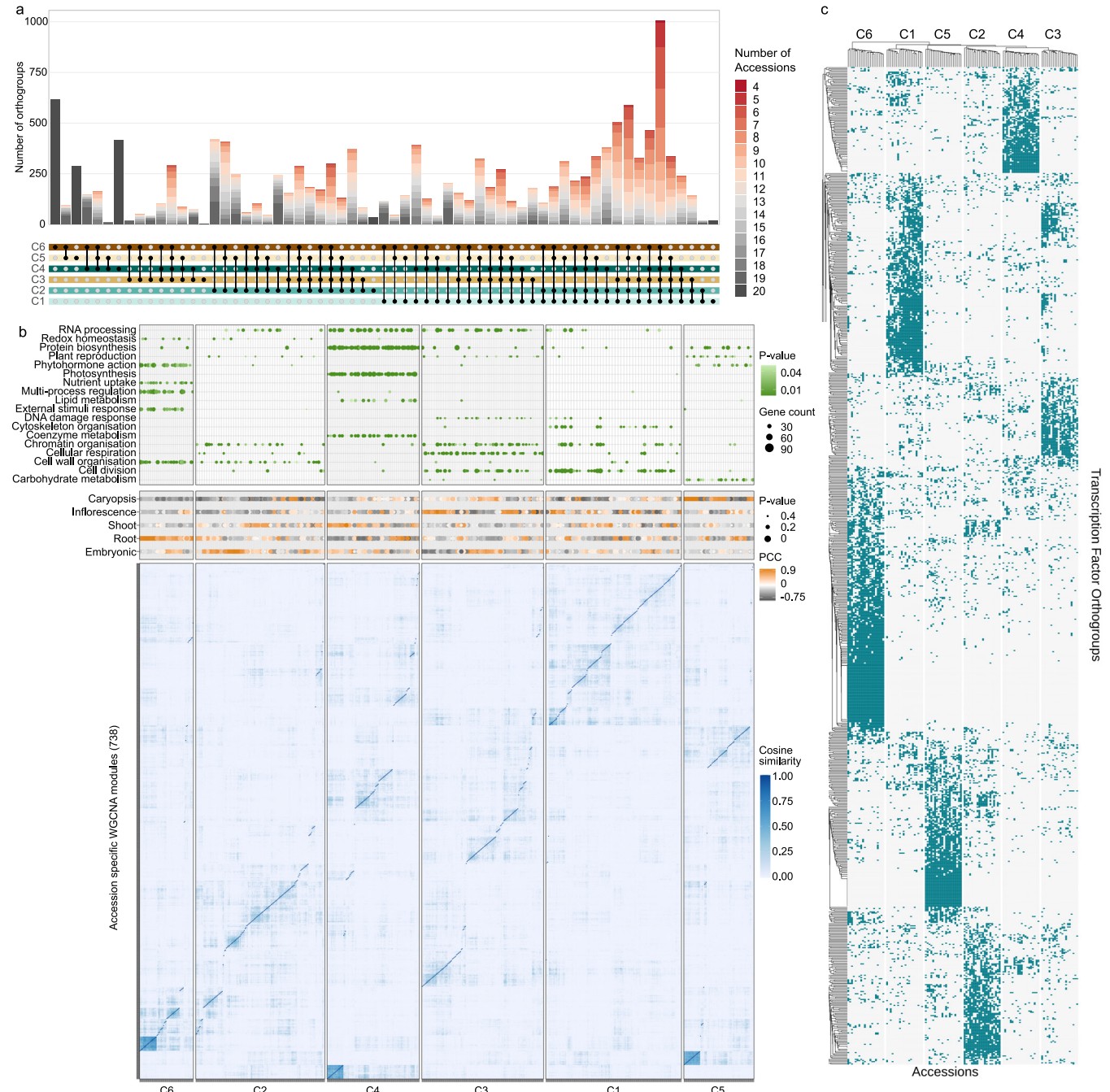

**Extended Data Fig. 5 | Community clustering. a**, Full distribution of 1:1 orthologous groups (exactly 20 genes, one from each genotype) across the six communities. Number of accessions refers to the largest cluster within a single community for the respective orthogroup. **b**, Heatmap displaying the Louvain community clustering of all 738 WGCNA modules (bottom). Six main communities were identified (x axis C1–C6), which show distinct functional patterns where the color of the dot represents the false discovery rate (FDR) adjusted p value of one-sided Fisher's exact test (top) and module-tissue correlations where the size of the dot represents the p value of two-sided Fisher's exact test (middle). **c**, The 861 transcription factors presence–absence clustering representation across the six communities.

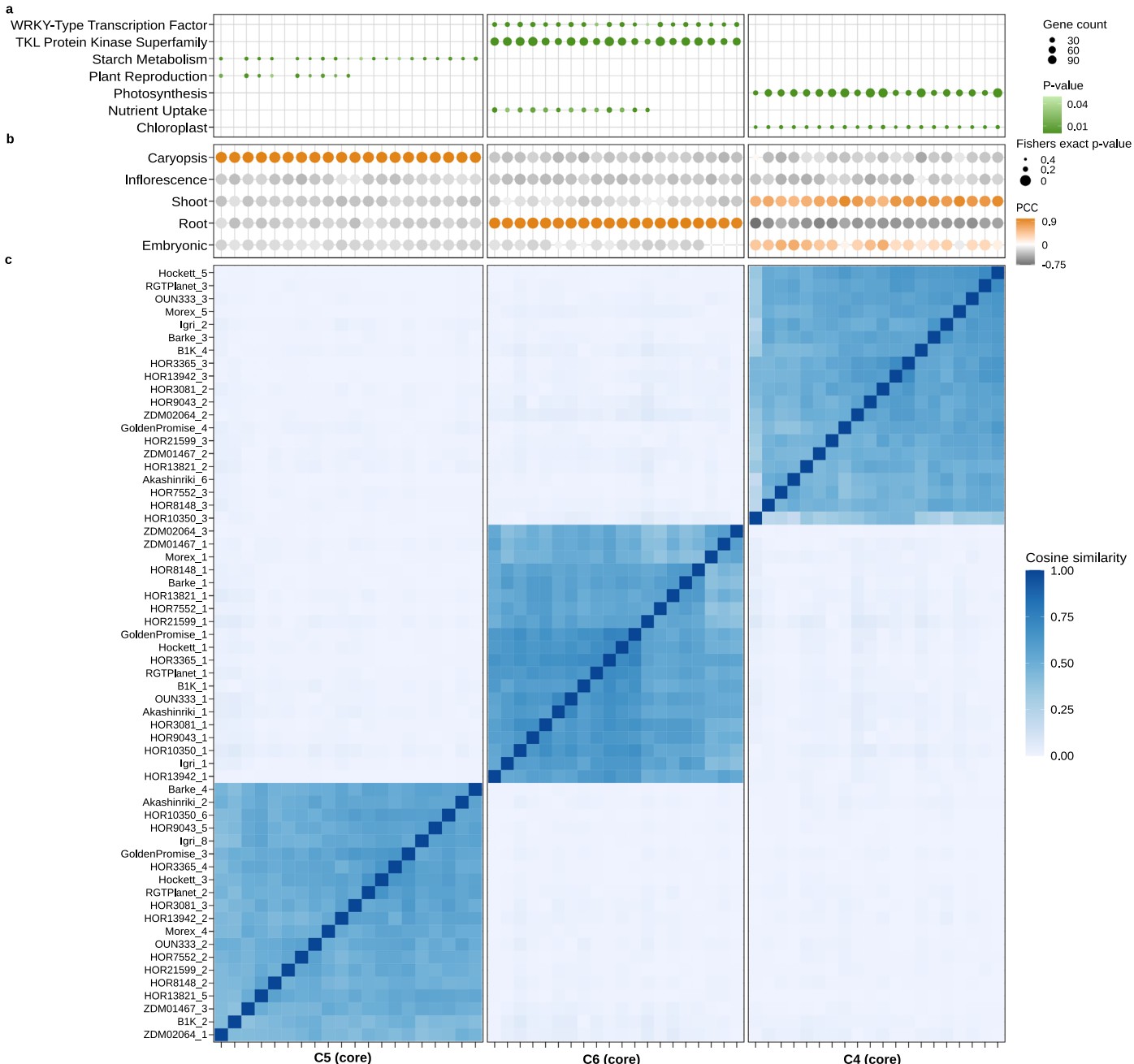

**Extended Data Fig. 6 | Functional enrichment, tissue specificity and correlations among the three most connected modules. a**, Summary of functional enrichments where the color of the dot represents the false discovery rate (FDR) adjusted p value of one-sided Fisher's exact test. **b**, Module-tissue correlations where the size of the dot represents the p value of two-sided Fisher's exact test for (**c**) the 3 most connected groups (C4, C5 and C6) of modules with cosine similarity cutoff 0.43 (zoom-in of Extended Data Fig. 4) displaying their cosine similarity values and accession wise module names.

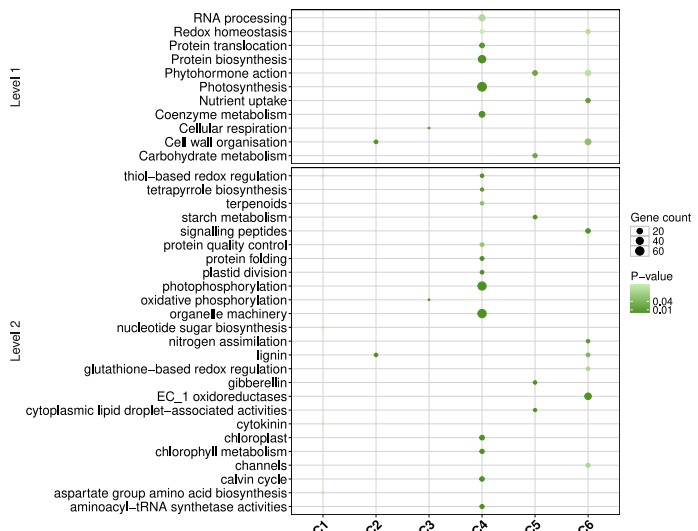

**Extended Data Fig. 7 | Functional enrichment within one community.** Enrichment for orthologous groups exclusively present in one community for the two highest hierarchical Mercator categories (level 1 and level 2) for 17 orthologous groups in C1, 34 in C2, 1 in C3, 414 in C4, 288 in C5 and 616 in C6. The color of the dot represents the false discovery rate (FDR) adjusted p value of one-sided Fisher's exact test.

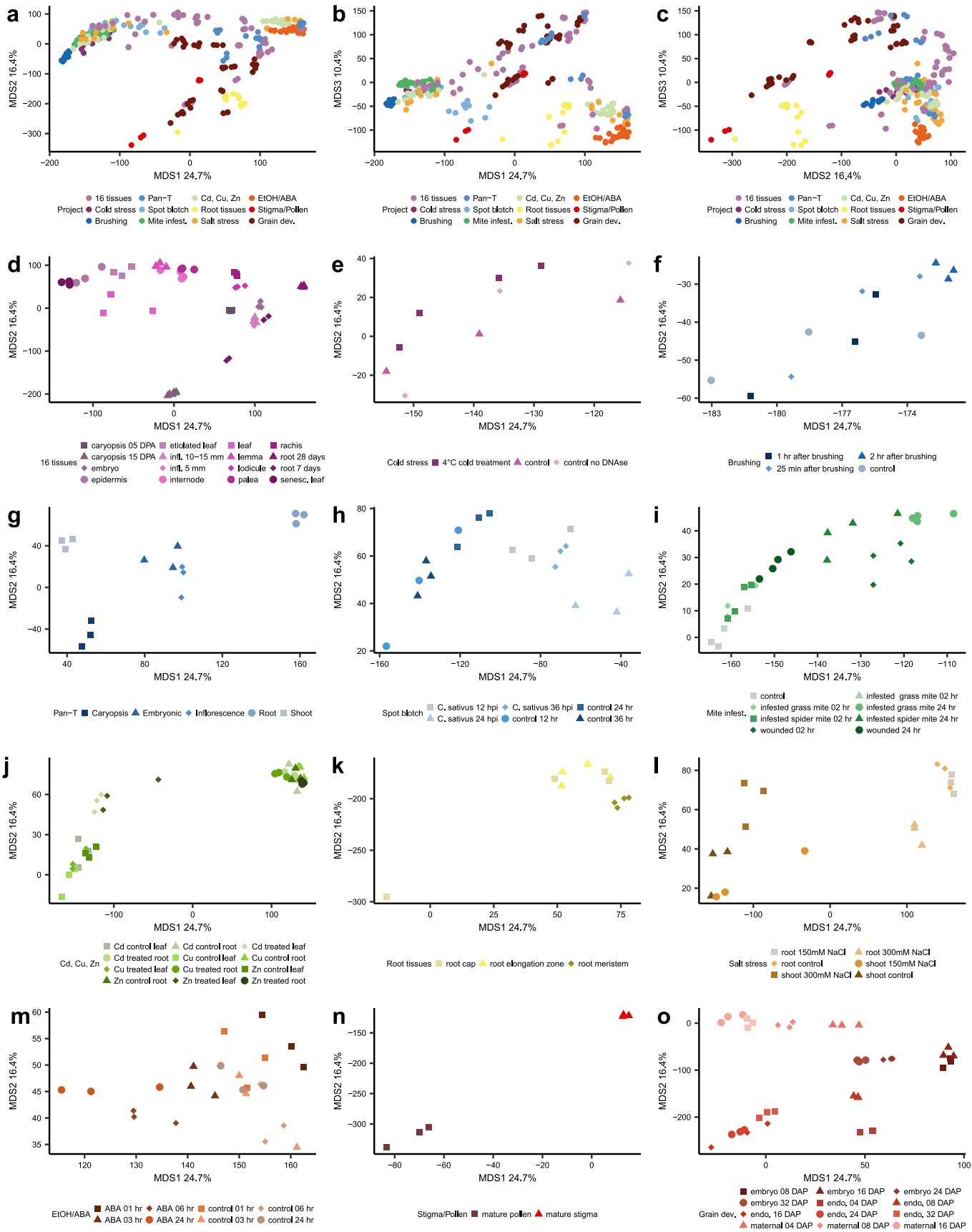

**Extended Data Fig. 8 | Multidimensional scaling plots of gene expression of the 12 RNA-seq datasets used to generate the Morex Gene Atlas. a–c,** First three dimensions show the distribution of samples from all projects. **d–o,** First two dimensions of the gene expression data from the individual projects.

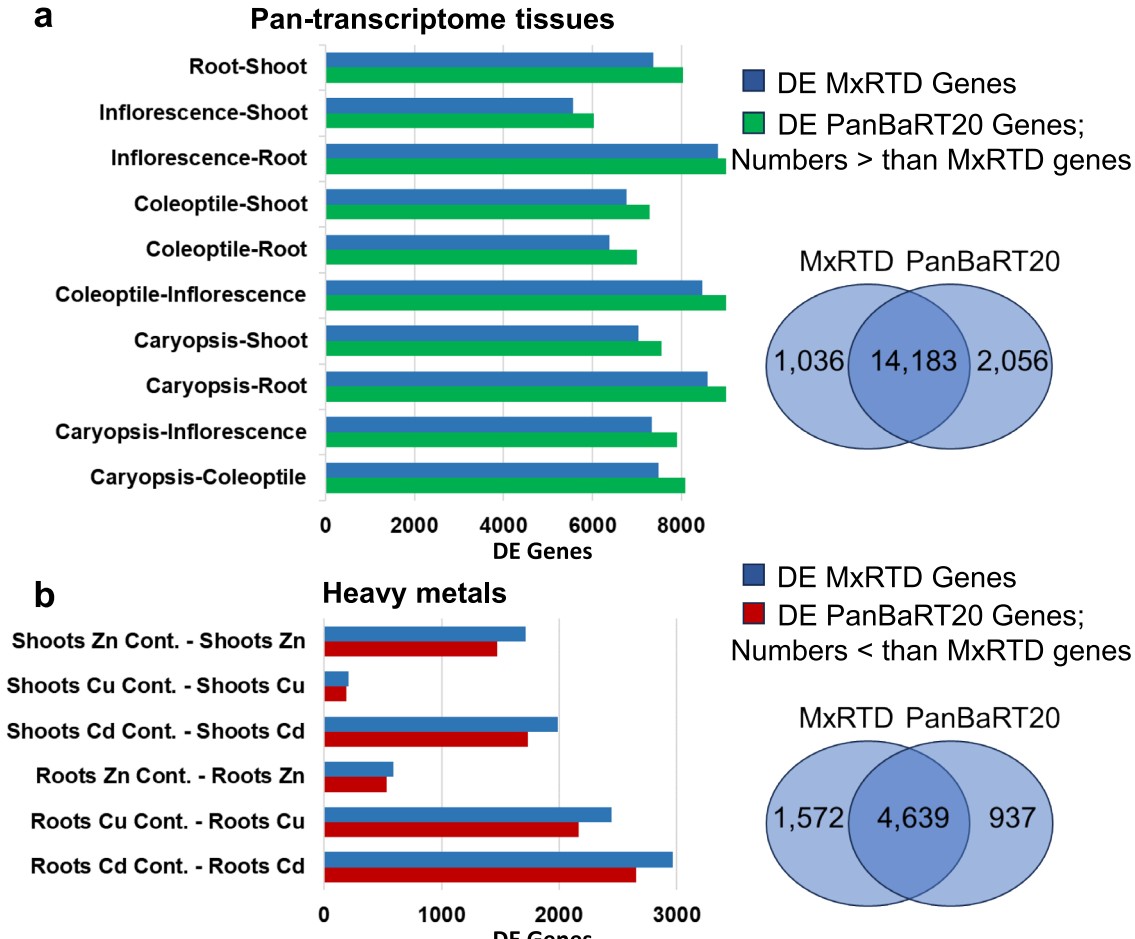

**Extended Data Fig. 9 | Differentially expressed genes (DEGs) in two experimental datasets. a**, Results from the pan-transcriptome sample dataset (PRJEB64639). **b**, Results from the heavy metal experimental dataset (PRJNA382490). Graphs show the numbers of DEGs in each named contrast group. Samples were pre-processed with cutoff of 10 CPM in a least two of the different samples. DEGs were selected with a greater than 2-fold change and a significance value of p ≤ 0.01. Statistics were performed using the limma-voom R package with multiple comparison adjustments using the Benjamini–Hochberg procedure. The blue box represents the total number of DEGs calculated using the Morex RTD (MxRTD), and the green and red boxes represent the total number of DEGs calculated using the PanBaRT20 RTD. Green represents PanBaRT20, which showed more DEGs compared to MxRTD, and the red represents PanBaRT20, which showed less DEGs compared to MxRTD. The Venn diagrams show the total number of unique genes across all the contrast groups tested using both MxRTD and PanBaRT20 RTD. The overlap represents the number of DEGs common to both RTDs.

# Reporting Summary

## Statistics

For all statistical analyses, confirm that the following items are present in the figure legend, table legend, main text, or Methods section.

| n/a | Confirmed | |
|---|---|---|
| ☐ | ☒ | The exact sample size (*n*) for each experimental group/condition, given as a discrete number and unit of measurement |
| ☐ | ☒ | A statement on whether measurements were taken from distinct samples or whether the same sample was measured repeatedly |
| ☐ | ☒ | The statistical test(s) used AND whether they are one- or two-sided *Only common tests should be described solely by name; describe more complex techniques in the Methods section.* |
| ☐ | ☒ | A description of all covariates tested |
| ☐ | ☒ | A description of any assumptions or corrections, such as tests of normality and adjustment for multiple comparisons |
| ☐ | ☒ | A full description of the statistical parameters including central tendency (e.g. means) or other basic estimates (e.g. regression coefficient) AND variation (e.g. standard deviation) or associated estimates of uncertainty (e.g. confidence intervals) |
| ☐ | ☒ | For null hypothesis testing, the test statistic (e.g. *F*, *t*, *r*) with confidence intervals, effect sizes, degrees of freedom and *P* value noted *Give P values as exact values whenever suitable.* |
| ☒ | ☐ | For Bayesian analysis, information on the choice of priors and Markov chain Monte Carlo settings |
| ☒ | ☐ | For hierarchical and complex designs, identification of the appropriate level for tests and full reporting of outcomes |
| ☒ | ☐ | Estimates of effect sizes (e.g. Cohen's *d*, Pearson's *r*), indicating how they were calculated |

*Our web collection on statistics for biologists contains articles on many of the points above.*

## Software and code

Policy information about availability of computer code

| Data collection | No software were used for data collection. |
|---|---|
| Data analysis | Multiple published software packages were used in the analyses and are described at https://github.com/cropgeeks/barleyPantranscriptome. Specific customized scripts (e.g. for gene cluster filtering) are included in a common folder at https://github.com/cropgeeks/barleyPantranscriptome/tree/main/scripts. Tools and software was used in the data analysis include: Fastp v0.20.1, STAR v2.7.8a, Stringtie v2.1.5, Scallop v0.10.5, IsoSeqv3 v3.4.0, TAMA (Dec 24th, 2022), PSVCP v1.0.1, Minimap2 v2.24, Transuite v0.2.2, InterProScan v5.59-91.0, Salmon v1.4.0, 3D RNA-seq v2.0.1, CD-HIT v4.8.1 , BLAST 2.15.0, SyRI v1.6.3, plotsr v0.5.4, Polyester v1.29.1, DESeq2 v1.34.0, WGCNA v1.69, Mercator4 v5.0, clusterProfiler v4.6, Python v3.10.12, networkx v3.1, Netgraph v4.12.11, OrthoFinder v2.5.5, RTDmaker v0.1.5, OmicsBox v3.1.2 and seqPattern v1.32 |

For manuscripts utilizing custom algorithms or software that are central to the research but not yet described in published literature, software must be made available to editors and reviewers. We strongly encourage code deposition in a community repository (e.g. GitHub). See the Nature Portfolio guidelines for submitting code & software for further information.

## Data

Policy information about availability of data

All manuscripts must include a data availability statement. This statement should provide the following information, where applicable:
- Accession codes, unique identifiers, or web links for publicly available datasets
- A description of any restrictions on data availability
- For clinical datasets or third party data, please ensure that the statement adheres to our policy

All raw data is available through the European Nucleotide Archive (https://www.ebi.ac.uk/ena/browser/home). Raw Illumina reads are available as BioProject accession PRJEB64639 (https://www.ebi.ac.uk/ena/browser/view/PRJEB64639), Iso-Seq CCS reads are available as BioProject accession PRJEB64637 (https://www.ebi.ac.uk/ena/browser/view/PRJEB64637). All data underpinning PanBaRT20 and MorexGeneAtlas are also available in Eorna (https://ics.hutton.ac.uk/panbart20/index.html) and (https://ics.hutton.ac.uk/morexgeneatlas/index.html). The Morex V3 pseudomolecules are available at http://doi.org/10.5447/ipk/2021/3. All supplementary data files have been uploaded to Figshare repositories and can be accessed here: https://figshare.com/s/1a6d27f41385efe5a790?file=47048692, https://figshare.com/s/1a6d27f41385efe5a790?file=47048698 https://figshare.com/s/1a6d27f41385efe5a790?file=47048704

## Research involving human participants, their data, or biological material

Policy information about studies with human participants or human data. See also policy information about sex, gender (identity/presentation), and sexual orientation and race, ethnicity and racism.

| | |
|---|---|
| Reporting on sex and gender | n/a |
| Reporting on race, ethnicity, or other socially relevant groupings | n/a |
| Population characteristics | n/a |
| Recruitment | n/a |
| Ethics oversight | n/a |

Note that full information on the approval of the study protocol must also be provided in the manuscript.

# Field-specific reporting

Please select the one below that is the best fit for your research. If you are not sure, read the appropriate sections before making your selection.

☒ Life sciences ☐ Behavioural & social sciences ☐ Ecological, evolutionary & environmental sciences

For a reference copy of the document with all sections, see nature.com/documents/nr-reporting-summary-flat.pdf

# Life sciences study design

All studies must disclose on these points even when the disclosure is negative.

| | |
|---|---|
| Sample size | The 20 barley pan-genome genotypes8 were used to prepare a barley pan-transcriptome. These are Akashinriki, Barke, ZDM02064, ZDM01467, B1K-04-12, GoldenPromise, Hockett, HOR10350, HOR13821, HOR13942, HOR21599, HOR3081, HOR3365, HOR7552, HOR8148, HOR9043, Igri, Morex, OUN333 and RGT Planet. Five different tissue/organ types including embryo, mesocotyl and seminal roots (together referred to throughout as embryonic tissue), seedling shoot, seedling root, inflorescence and caryopsis were collected in three biological replicates amounting to 300 samples. |
| Data exclusions | Pan Transcriptome RNA-seq data: 2 samples (HOR10350 Caryopsis rep3 and HOR7552 Root rep2) were excluded due to poor RNA quality. Hor8148_In3 was identified as mislabelled thus also being excluded.<br>CNV analyses: Gene clusters with a median expression below 10 TPM were excluded from association analyses between CNV and expression. Gene network analyses: Out of 13,680 orthologous single copy genes, 13,652 genes passed the filter of lowly expressed genes (TPM > 0.5 in minimum 2 biological replicates) in all accessions and were propagated to the following analysis. From the original 297 samples, principal component analysis (PCA) revealed that one sample (ZDM01467_In2) was a severe outlier, therefore, it was removed from further network analysis.<br>For differential expression analysis from the Morex atlas: Transcripts determined as expressed in ≤ 2 of the samples with count per million reads (CPM) ≤ 10 were removed from further analysis.<br>All data exclusions are described in the manuscript 'Main Text' and/or 'Methods' |
| Replication | Three biological replicates per sample except 2 samples excluded due to poor RNA quality and one sample identified as mislabelled as described above |
| Randomization | No randomization is required as the plant background and condition is exactly the same. There is no biases on how we choose the plants or tissues. |

| Blinding | No blinding is required as the there is no participant's biases involved in this plant study |
|---|---|

# Reporting for specific materials, systems and methods

We require information from authors about some types of materials, experimental systems and methods used in many studies. Here, indicate whether each material, system or method listed is relevant to your study. If you are not sure if a list item applies to your research, read the appropriate section before selecting a response.

## Materials & experimental systems

| n/a | Involved in the study |
|---|---|
| ☒ | Antibodies |
| ☒ | Eukaryotic cell lines |
| ☒ | Palaeontology and archaeology |
| ☒ | Animals and other organisms |
| ☒ | Clinical data |
| ☒ | Dual use research of concern |
| ☐ | ☒ Plants |

## Methods

| n/a | Involved in the study |
|---|---|
| ☒ | ChIP-seq |
| ☒ | Flow cytometry |
| ☒ | MRI-based neuroimaging |

## Dual use research of concern

Policy information about dual use research of concern

### Hazards

Could the accidental, deliberate or reckless misuse of agents or technologies generated in the work, or the application of information presented in the manuscript, pose a threat to:

| No | Yes | |
|---|---|---|
| ☒ | ☐ | Public health |
| ☒ | ☐ | National security |
| ☒ | ☐ | Crops and/or livestock |
| ☒ | ☐ | Ecosystems |
| ☒ | ☐ | Any other significant area |

### Experiments of concern

Does the work involve any of these experiments of concern:

| No | Yes | |
|---|---|---|
| ☒ | ☐ | Demonstrate how to render a vaccine ineffective |
| ☒ | ☐ | Confer resistance to therapeutically useful antibiotics or antiviral agents |
| ☒ | ☐ | Enhance the virulence of a pathogen or render a nonpathogen virulent |
| ☒ | ☐ | Increase transmissibility of a pathogen |
| ☒ | ☐ | Alter the host range of a pathogen |
| ☒ | ☐ | Enable evasion of diagnostic/detection modalities |
| ☒ | ☐ | Enable the weaponization of a biological agent or toxin |
| ☒ | ☐ | Any other potentially harmful combination of experiments and agents |

## Plants

| | |
|---|---|
| Seed stocks | Inbred seed stocks for the pan-transcriptome study were obtained from IPK-Gatersleben as described in Supplementary Table 2 and Jayakodi, M. et al. (2020 and 2023) and are available from German federal ex situ genebank at IPK Gatersleben.de.  For the 'inversion study' seed were obtained from the JHI 'seed store' and are as described in Schreiber, M. et al (2023)  https://doi:10.1101/2023.03.06.531259 and listed in Supplementary table 7 and are available from the authors on request. |
| Novel plant genotypes | n/a |
| Authentication | Genotypes for pan-transcriptome studies were verified by comparison to SNP polymorphism data derived from the same genotypes used in the barley pan-genome studies of Jayakodi et al.  The same inbred seed stocks were used for both studies.  For the '141Mb inversion' study all genotypes were verified by comparing RNAseq derived SNPs to barley 50K SNP chip data deposited in a reference 'Germinate' database.  For the 'Morex atlas' study all sequences were labeled 'cv. Morex' in the SRA and were assumed to be genuine cv. Morex genotypes. |

