## [Peer Review File · Nature Genetics]

A barley pan-transcriptome reveals layers of genotype-dependent transcriptional complexity

Corresponding Author: Professor Robbie Waugh

This manuscript has been previously reviewed at another journal. This document only contains information relating to versions considered at Nature Genetics.

Version 0:

Decision Letter:

Our ref: NG-A65801-T

28th Aug 2024

Dear Dr. Waugh,

Thank you for submitting your revised manuscript "A barley pan-transcriptome reveals layers of genotype-dependent transcriptional complexity" (NG-A65801-T). It has now been seen by the original referees and their comments are below. The reviewers find that the paper has improved in revision, and therefore we'll be happy in principle to publish it in Nature Genetics, pending minor revisions to satisfy the referees' final requests and to comply with our editorial and formatting guidelines.

Sincerely,

Wei Li, PhD
Senior Editor
Nature Genetics
www.nature.com/ng

Reviewer #1 (Remarks to the Author):

I have reviewed the latest revisions and the authors' detailed responses to my previous comments. All of my questions and concerns have been addressed, and I have no further comments.

The resources, datasets, and results are well presented and of interest to the barley community. However, I defer to the editors to determine whether this manuscript falls within the journal's scope, as I still believe it is mainly descriptive, despite the addition of the new paragraph "Data use scenario" which for me, lacks validation of the main results.

Reviewer #3 (Remarks to the Author):

The authors did a nice job addressing my concerns from the original submission.

The only additional item I found is that the expression atlas websites require a username and password to enter, so I was unable to review its content.

<https://ics.hutton.ac.uk/morexgeneatlas/index.html>
